# FedLPA: Personalized One-shot Federated Learning with Layer-Wise Posterior Aggregation

## Abstract

Efficiently aggregating trained neural networks from local clients into a global model on a server is a widely researched topic in federated learning. Recently, motivated by diminishing privacy concerns, mitigating potential attacks, and reducing the overhead of communication, one-shot federated learning (i.e., limiting client-server communication into a single round) has gained popularity among researchers. However, the one-shot aggregation performances are sensitively affected by the non-identical training data distribution, which exhibits high statistical heterogeneity in some real-world scenarios. To address this issue, we propose a novel one-shot aggregation method with Layer-wise Posterior Aggregation, named FedLPA. FedLPA aggregates local models to obtain a more accurate global model without requiring extra auxiliary datasets or exposing any confidential local information, e.g., label distributions. To effectively capture the statistics maintained in the biased local datasets in the practical non-IID scenario, we efficiently infer the posteriors of each layer in each local model using layer-wise Laplace approximation and aggregate them to train the global parameters. Extensive experimental results demonstrate that FedLPA significantly improves learning performance over state-of-the-art methods across several metrics.

## 1 Introduction

The significance of data privacy in Deep Learning (LeCun et al., 2015; Schmidhuber, 2015; Zhang et al., 2018; Krizhevsky et al., 2017; Amodei et al., 2016; Pouyanfar et al., 2018b;a) has surged to the forefront as a major global concern (Yang et al., 2019). With the primary objectives of safeguarding data privacy and curbing the aggregation and management of data across institutions, the distribution of data exhibits variations among clients (Yang et al., 2019). In the burgeoning domain of machine learning, federated learning (FL), as denoted by references (McMahan et al., 2016; Kairouz et al., 2021; Li et al., 2021), has emerged as a prominent paradigm. The fundamental tenet of federated learning revolves around the sharing of machine learning models derived from decentralized data repositories, as opposed to divulging the raw data itself. This approach effectively preserves the confidentiality of individual data.

The standard federated learning framework, Fedavg (McMahan et al., 2016; 2017), involves local model training and aggregating these local models into a global one through parameter averaging. However, the majority of current FL algorithms like Fedavg necessitate numerous communication rounds to effectively train a global model. This leads to substantial communication overhead, heightened privacy concerns, and heightened demands for fault tolerance throughout the rounds. One-shot FL, which limits client-server communication into one round as explored in previous works (Guha et al., 2019; Li et al., 2020a; Diao et al., 2023), has emerged as a promising yet challenging scheme to address these issues. It proves particularly practical in such scenarios where iterative communication is not feasible. Additionally, a reduction in communication rounds translates to fewer opportunities for any potential eavesdropping attacks.

While one-shot FL shows promise, existing methods often grapple with several challenges such as non-independent and non-identically distributed (non-IID) data (Zhou et al., 2020; Zhang et al., 2021), or inadequate handling of high statistical heterogeneity information in the previous works

(Liu et al., 2021; Al-Shedivat et al., 2020). Moreover, some methods rely on an auxiliary public dataset to achieve satisfactory performance in one-shot FL (Guha et al., 2019; Li et al., 2020a), or even on pre-trained large models (Yang et al., 2023), which may not be practical (Zhu et al., 2021) in some sensitive scenarios. Additionally, certain approaches (Shin et al., 2020; Zhang et al., 2021; Heinbaugh et al., 2022; Diao et al., 2023) might expose data/label privacy to the local and global models, e.g., the client label distribution, potentially violating General Data Protection Regulation (GDPR) rules. Furthermore, some of these methods (Li et al., 2020a; Zhou et al., 2020; Zhang et al., 2022a) may require substantial computing resources for dataset distillation, model distillation, or even training a generator capable of generating synthetic data for second-stage training on the server side.

On the other hand, the one-shot FL performance always falls short when dealing with non-IID data. Non-IID data biases global updates, reducing the accuracy of the global model and slowing down convergence. In extreme non-IID cases, clients may be required to address distinct classes. To tackle this heterogeneity among clients, personalized federated learning (PFL) (Smith et al., 2017) in multi-round settings becomes essential, allowing each client to use a personalized model instead of a shared global model. With the personalized approach, the multi-round framework benefits from joint training while allowing each client to keep its own unique model. However, one-shot aggregation on the local model is far from being resolved under the personalized setting.

In this paper, we introduce a novel one-shot aggregation approach to address these issues, named FedLPA, i.e., federated learning with Layer-wise Posterior Aggregation. FedLPA infers the posteriors of each layer in each local model using the empirical Fisher information matrix obtained by Layer-wise Laplace Approximation. Laplace Approximations are widely used to compute the empirical Fisher information matrix for the neural networks, conveying the data statistics in personalized settings. However, computing empirical Fisher information matrices of multiple personalized local clients and aggregating their Fisher information matrices remains an ongoing challenge (Liu et al., 2021). FedLPA aggregates the posteriors of local models using the accurately computed block-diagonal empirical Fisher information matrices as a metric of the parameter space. This matrix captures essential parameter correlations and distinguishes itself from prior methods by being non-diagonal and non-low-rank, thereby conveying the statistics of biased local datasets. In our approach, we directly train global model parameters after the aggregation without any need for server-side knowledge distillation (Lin et al., 2020).

Our experiments verify the efficiency and effectiveness of FedLPA, highlighting that FedLPA markedly enhances the test accuracy when compared to existing one-shot FL approaches across various datasets. Our main contributions are summarized as follows:

• To the best of our knowledge, we are the first to propose a personalized one-shot federated learning approach that directly trains the global models using the block-diagonal empirical Fisher information matrices. Our approach is data-free without the need for any auxiliary information and significantly enhances the system performance, including negligible communication cost and moderate computing overhead.

• We are the first to train the global model parameters via constructing a multi-variates linear objective function and using its quadratic form, which allows us to formulate and solve this problem in a convex form. Nevertheless, from the theoretical analysis, we show that FedLPA has a linear convergence rate, ensuring good performance.

• We conduct extensive experiments to illustrate the effectiveness of FedLPA. Our approach consistently outperforms the baselines, showcasing substantial enhancements across various settings and datasets. Even in some extreme scenarios with severe label skew, e.g., where each client has only one class, in which many federated learning algorithms struggle, we achieve satisfactory results.

## 2 BACKGROUND AND RELATED WORKS

### 2.1 FEDERATED LEARNING

Previous work Fedavg (McMahan et al., 2016) first introduced the concept of FL and presented the algorithm, which achieved competitive performance on i.i.d data, in comparison to several centralized techniques. However, it was observed in previous works (Li et al., 2019; Zhao et al., 2018) that

the convergence rate and ultimate accuracy of Fedavg on non-IID data distributions were significantly reduced, compared to the results observed with homogeneous data distributions.

Other methods have been developed to enhance performance in federated learning. The SCAFFOLD method (Karimireddy et al., 2020) leveraged control variates to reduce objective inconsistency in local updates. It estimated the drift of directions in local optimization and global optimization and incorporated this drift into local training to align the local optimization direction with the global optimization. Fednova (Wang et al., 2020b) addressed objective inconsistency while maintaining rapid error convergence through a normalized averaging method. It scaled and normalized the local updates of each client based on the number of local optimization steps. Fedprox (Li et al., 2020c) enhanced the local training process by introducing a global prior in the form of an $L2$ regularization term within the local objective function. In Yurochkin et al. (2019); Wang et al. (2020a), researchers introduced PFNM, a Bayesian probabilistic framework specifically tailored for multilayer perceptrons. PFNM employed a Beta-Bernoulli process (BBP) (Thibaux & Jordan, 2007) to aggregate local models, quantifying the degree of alignment between global and local parameters. The framework proposed in Liu et al. (2021) utilized a multivariate Gaussian product method to construct a global posterior by aggregating local posteriors estimated using an online Laplace approximation. FedPA (Al-Shedivat et al., 2020) also applied the Gaussian product method but employed stochastic gradient Markov chain Monte Carlo for approximate inference of local posteriors. DAFL (Data-Free Learning) (Chen et al., 2019) introduced an innovative framework based on generative adversarial networks. ADI (Yin et al., 2020) utilized an image synthesis method that leveraged the image distribution to train deep neural networks without real data. The pFedHN method (Shamsian et al., 2021) incorporated HyperNetworks (Krueger et al., 2017) to address federated learning applications.

However, all of these methods encountered challenges in the personalized one-shot federated learning setting, as they required aggregating the model by multiple rounds and might be inaccurate due to the omission of critical information, such as posterior joint probabilities between different parameters.

## 2.2 ONE-SHOT FEDERATED LEARNING

One-shot Federated Learning (FL) is an emerging and promising research direction characterized by its minimal communication cost. In the first study on one-shot FL (Guha et al., 2019), the approach involved on the aggregation of local models, forming an ensemble to construct the final global model. Subsequently, knowledge distillation using public data was applied in the following step. FedKT (Li et al., 2020a) brought forward the concept of consistent voting to fortify the ensemble. Recent research endeavors (Zhang et al., 2021; 2022a) proposed data-free knowledge distillation schemes tailored for one-shot FL. These methods adopted the basic ensemble distillation framework as FedDF (Lin et al., 2020). XorMixFL (Shin et al., 2020) introduced the use of exclusive OR operation (XOR) for encoding and decoding samples in data sharing. It is important to note that XorMixFL assumed the possession of labeled samples from a global class by all clients and the server, which might not align with practical real-world scenarios. A noteworthy innovation of DENSE (Zhang et al., 2022a) was its utilization of a generator to create synthetic datasets on the server side, circumventing the need for a public dataset in the distillation process. FedOV (Diao et al., 2023) delved into addressing comprehensive label skew cases. FEDCVAE (Heinbaugh et al., 2022) confronted this challenge by transmitting all label distributions from clients to servers. These schemes (Shin et al., 2020; Li et al., 2020a; Zhang et al., 2021; 2022a; Heinbaugh et al., 2022; Diao et al., 2023) exposed some client-side private information, leading to additional communication overhead and potential privacy leakage, e.g., FEDCVAE (Heinbaugh et al., 2022) needed all the client label distribution to be transmitted to the server side and FedOV Diao et al. (2023) needed the clients to know the labels which were unknown. Instead, MA-Echo (Su et al., 2023) adopted a unique approach by emphasizing the addition of norms among layer-wide parameters during the aggregation of local models. FedFisher (Jhunjhunwala et al., 2023) also leveraged empirical Fisher information matrix, but focused on theoretic analysis of the error on its approximation method. However, their method grappled with limited experiments and lacked detailed explanations of the approach. FedDISC (Yang et al., 2023), on the other hand, relied on the pre-trained model CLIP from OpenAI, where their reliance might not always align with practicality or suitability for diverse scenarios.

While some of these techniques are orthogonal to FedLPA and can be integrated with it, it is worth noting that none of the previously mentioned algorithms possess the capability to train global model parameters using empirical Fisher information matrices. Some of them may require additional information, entail the risk of breaching data/label privacy.

## 3 METHODOLOGY

### 3.1 OBJECTIVE FORMULATION

Generally, federated learning is defined as a optimization problem (Li et al., 2020b;c; Karimireddy et al., 2020; Wang et al., 2020b) for maximizing a global objective function $\mathbb{F}(\boldsymbol{\theta})$ which is a mixture of local objective functions $\mathbb{F}_k(\boldsymbol{\theta}, \mathcal{D}_k)$:

$$\mathbb{F}(\boldsymbol{\theta}) = \sum_{k=1}^{K} \mathbb{F}_k(\boldsymbol{\theta}, \mathcal{D}_k),$$ (1)

where $\boldsymbol{\theta} = [\text{vec}(\mathbf{W}_1), \ldots, \text{vec}(\mathbf{W}_l), \ldots, \text{vec}(\mathbf{W}_L)]$ is the parameter vector of global model and $\mathbf{W}_l$ is the weight and bias of layer $l$ for a $L$-layers neural network; $\mathcal{D}_k$ is the local dataset $k$-th client. $\mathbb{F}_k(\boldsymbol{\theta}, \mathcal{D}_k)$ is the expectation of the local objective function, which is proportional to the logarithm of likelihood $\log p(\mathcal{D}_k|\boldsymbol{\theta})$.

Previous works (Liu et al., 2021; Al-Shedivat et al., 2020) give a common formula of the global posterior which consists of local posteriors $p(\boldsymbol{\theta}|\mathcal{D}_k)$ under variational inference formulation:

$$p(\boldsymbol{\theta}|\mathcal{D}) \propto \prod_{k=1}^{K} p(\mathcal{D}_k|\boldsymbol{\theta}) \propto \prod_{k=1}^{K} p(\boldsymbol{\theta}|\mathcal{D}_k).$$ (2)

$$\max_{\boldsymbol{\theta}} \mathbb{F}(\boldsymbol{\theta}) = \sum_{k=1}^{K} \frac{|D_k|}{|\mathcal{D}|} \cdot \mathbb{E}_{s \in \mathcal{D}_k} [\log p(s|\boldsymbol{\theta})] \equiv \max_{\boldsymbol{\theta}} \prod_{k=1}^{K} p(\boldsymbol{\theta}|\mathcal{D}_k).$$ (3)

As we know, the objective function is the expectation of the likelihood, and the sum of the logarithms is equal to the logarithms of the product as Eq. 3. Therefore, globally variational inference using Eq. 2 is equivalent to optimization for Eq. 1. Correspondingly, we get:

$$\max_{\boldsymbol{\theta}} \mathbb{F}_k(\boldsymbol{\theta}, \mathcal{D}_k) \equiv \max_{\boldsymbol{\theta}} p(\boldsymbol{\theta}|\mathcal{D}_k).$$ (4)

Following the same training pattern of federated learning, each client infers the local posterior $p(\boldsymbol{\theta}|\mathcal{D}_k)$ by using the local dataset $\mathcal{D}_k$, and then uploads probability parameters to the server. As a result, the server obtains the global posterior $p(\boldsymbol{\theta}|\mathcal{D})$ by aggregating local posteriors using Eq. 2.

However, both the global and local posterior are usually intractable because modern neural networks are usually non-linear and have a large number of parameters. Therefore, it is necessary to design an efficient and accurate aggregation method for one-shot federated learning.

### 3.2 APPROXIMATING POSTERIORS

Although the posterior is usually intractable, the posterior can be approximated as a Gaussian distribution by performing a Taylor expansion on the logarithm of the posterior (Ritter et al., 2018):

$$\log p(\boldsymbol{\theta}|\mathcal{D}) \approx \log p(\boldsymbol{\theta}^*|\mathcal{D}) - \frac{1}{2}(\boldsymbol{\theta} - \boldsymbol{\theta}^*)^\top \bar{\mathbf{H}}(\boldsymbol{\theta} - \boldsymbol{\theta}^*),$$ (5)

where $\boldsymbol{\theta}^*$ is the optimal parameter vector, $\bar{\mathbf{H}} = \mathbb{E}_{s \in \mathcal{D}}[\mathbf{H}]$ is the average Hessian of the negative log posterior over a dataset $\mathcal{D}$. It is reasonable to approximate global and local posteriors as multivariates Gaussian distributions with expectations $\bar{\boldsymbol{\mu}} = \boldsymbol{\theta}^*$ and $\boldsymbol{\mu_k} = \boldsymbol{\theta}_k^*$; co-variances $\bar{\boldsymbol{\Sigma}} = \bar{\mathbf{H}}^{-1}$ and $\boldsymbol{\Sigma}_k = \bar{\mathbf{H}}_k^{-1}$ (Daxberger et al., 2021). The details are discussed in Appendix B.

$$p(\boldsymbol{\theta}|\mathcal{D}) \equiv \boldsymbol{\theta} \sim \mathcal{N}(\bar{\boldsymbol{\mu}}, \bar{\boldsymbol{\Sigma}}), \quad p(\boldsymbol{\theta}|\mathcal{D}_k) \equiv \boldsymbol{\theta} \sim \mathcal{N}(\boldsymbol{\mu_k}, \boldsymbol{\Sigma}_k).$$ (6)

As a result, if given local expectation $\boldsymbol{\mu}_k$ and local co-variance $\boldsymbol{\Sigma}_k$, the global posterior is determined by Eq. 2 as below:

$$\bar{\boldsymbol{\mu}} = \bar{\boldsymbol{\Sigma}} \sum_k^K \boldsymbol{\Sigma}_k^{-1} \boldsymbol{\mu}_k, \quad \bar{\boldsymbol{\Sigma}}^{-1} = \sum_k^K \boldsymbol{\Sigma}_k^{-1}. \tag{7}$$

Modern algorithms (Rumelhart et al., 1986; Martens & Grosse, 2015a) allow the local training process to obtain an optimal, regarded as the expectation $\boldsymbol{\mu}_k$ in the above equations. However, $\bar{\mathbf{H}}_k$ is intractable to compute due to a large number of parameters in modern neural networks. An efficient method is to approximate $\bar{\mathbf{H}}_k$ using the empirical Fisher information matrix (Van Loan, 2000).

### 3.3 INFERRING THE LOCAL LAYER-WISE POSTERIORS WITH THE BLOCK-DIAGONAL EMPIRICAL FISHER INFORMATION MATRICES

A empirical Fisher $\tilde{\mathbf{F}}$ is defined as below:

$$\tilde{\mathbf{F}} = \sum_{s \in \mathcal{D}} \left[ \nabla \log p(s|\boldsymbol{\theta}) \nabla \log p(s|\boldsymbol{\theta})^\top \right], \tag{8}$$

where $p(s|\boldsymbol{\theta})$ is the likelihood on data point $s$. It is an approximate of the Fisher information matrix, the empirical Fisher information matrix is equivalent to the expectation of the Hessian of the negative log posterior if assuming $p(s|\boldsymbol{\theta})$ is identical for each $s \in \mathcal{D}$.

Therefore, the local co-variance $\boldsymbol{\Sigma}_k$ can be approximated by the empirical Fisher $\tilde{\mathbf{F}}_k$ (Martens & Grosse, 2015b; Grosse & Martens, 2016). the details is discussed in Appendix C:

$$\boldsymbol{\Sigma}_k^{-1} \approx \tilde{\mathbf{F}}_k + \lambda \mathbf{I} \tag{9}$$

Kirkpatrick et al. (2017); Liu et al. (2021) ignore co-relations between different parameters and only consider the self-relations of parameters as computing all co-relations is impossible, which are inaccurate. In order to capture co-relations between different parameters efficiently, previous works (Martens & Grosse, 2015a; Ritter et al., 2018) estimate a block empirical Fisher information matrix $\mathbf{F}$ instead of assuming parameters are independent and approximating the co-variance by the diagonal of the empirical Fisher. As pointed out in Martens & Grosse (2015a); Benzing (2022); Zhang et al. (2022b), co-relations inner a layer are much more significant than others, while computing the co-relations between different layers brings slight improvement but much more computation. Therefore, assuming parameters are layer-independent is a good trade-off. As a result, the approximated layer-wise empirical Fisher is block-diagonal, while the details are shown in Appendix D. For layer $l$ on client $k$, its empirical Fisher $\mathbf{F}_{k_l}$ is one of the diagonal blocks in the whole empirical Fisher for the local model and is factored into two small matrices as below,

$$\boldsymbol{\Sigma}_{k_l}^{-1} \approx \mathbf{F}_{k_l} = \mathbf{A}_{k_l} \otimes \mathbf{B}_{k_l}, \tag{10}$$

where $\otimes$ is the Kronecker product; $\mathbf{A}_{k_l} = \hat{\mathbf{a}}_{k_{l-1}} \hat{\mathbf{a}}_{k_{l-1}}^\top + \pi_l \sqrt{\lambda} \mathbf{I}$ and $\mathbf{B}_{k_l} = \hat{\mathbf{b}}_{k_l} \hat{\mathbf{b}}_{k_l}^\top + \frac{1}{\pi_l} \sqrt{\lambda} \mathbf{I}$ are two factor matrices; $\hat{\mathbf{a}}_{k_l}$ is the activations and $\hat{\mathbf{b}}_{k_l}$ is the linear pre-activations of layer $l$ on client $k$, $\lambda$ is the hyperparameter and $\pi_l$ is a factor minimizing approximation error in $\mathbf{F}_{k_l}$ (Martens & Grosse, 2015a; Grosse & Martens, 2016; Botev et al., 2017). $\mathbf{A}_{k_l}$ and $\mathbf{B}_{k_l}$ are symmetric positive definite matrices (Rumelhart et al., 1986; Martens & Grosse, 2015a). The details are discussed in Appendix D.

We use $\boldsymbol{\theta}_{k_l}$ to denote the parameter vector of layer $l$ and $\boldsymbol{\mu}_{k_l} = \text{vec}(\mathbf{W}_{k_l}^*)$ is the vectorized optimal weight matrix of layer $l$ on client $k$. Thus, the resulting local layer-wise posterior approximation is $\boldsymbol{\theta}_{k_l} \sim \mathcal{N}(\boldsymbol{\mu}_{k_l}, \mathbf{F}_{k_l}^{-1})$.

## 3.4 ESTIMATING THE GLOBAL EXPECTATION

Given the local posteriors, the global expectation could be aggregated by Eq. 7. With Eq. 33, the $l$-th layer's global expectation $\bar{\boldsymbol{\mu}}_l$ consists of Kronecker products:

$$
\begin{aligned}
\bar{\boldsymbol{\mu}}_l &= \bar{\boldsymbol{\Sigma}}_l \sum_k^K \boldsymbol{\Sigma}_{k_l}^{-1} \boldsymbol{\mu}_{k_l} = \bar{\boldsymbol{\Sigma}}_l \sum_k^K (\mathbf{A}_{k_l} \otimes \mathbf{B}_{k_l}) \boldsymbol{\mu}_{k_l} \\
&= \bar{\boldsymbol{\Sigma}}_l \sum_k^K \operatorname{vec}(\mathbf{B}_{k_l} \mathbf{M}_{k_l} \mathbf{A}_{k_l}) = \bar{\boldsymbol{\Sigma}}_l \sum_k^K \mathbf{z}_{k_l} = \bar{\boldsymbol{\Sigma}}_l \bar{\mathbf{z}}_l,
\end{aligned}
\tag{11}
$$

where $\bar{\mathbf{z}}_l = \sum_k^K \mathbf{z}_{k_l}$ and $\mathbf{z}_{k_l} = \operatorname{vec}(\mathbf{B}_{k_l} \mathbf{M}_{k_l} \mathbf{A}_{k_l})$ is a immediate notations for simplification; $\mathbf{M}_{k_l}$ is the local expectation matrices that $\boldsymbol{\mu}_{k_l} = \operatorname{vec}(\mathbf{M}_{k_l})$. The corresponding global co-variance is an inverse of the sum of Kronecker products:

$$
\bar{\boldsymbol{\Sigma}}_l = (\sum_k^K \mathbf{A}_{k_l} \otimes \mathbf{B}_{k_l})^{-1}.
\tag{12}
$$

As shown in Eq. 11, obtaining the global expectation $\bar{\boldsymbol{\mu}}_l$ requires calculating the inverse of $\bar{\boldsymbol{\Sigma}}_l^{-1}$ as Eq. 12, which is unacceptable. Thus, we propose our method to directly train the parameters of the global model on the server side.

## 3.5 TRAIN THE PARAMETERS OF GLOBAL MODEL

Previous works (Martens & Grosse, 2015a; Grosse & Martens, 2016) approximate the expectation of Kronecker products by a Kronecker product of expectations $\mathbb{E}[\mathbf{A} \otimes \mathbf{B}] \approx \mathbb{E}[\mathbf{A}] \otimes \mathbb{E}[\mathbf{B}]$ with an assumption of $\mathbf{A}_{k_l}$ and $\mathbf{B}_{k_l}$ are independent, which is called Expectation Approximation (EA). However, it may lead to a biased global expectation. The details are discussed in Appendix E.

Instead, we could construct a linear objective after aggregating the approximation of local posteriors via using block-diagonal empirical fisher information matrices. We denotes $\bar{\mathbf{M}}$ as the matrix formula of $\bar{\boldsymbol{\mu}} = \operatorname{vec}(\bar{\mathbf{M}})$, and the optimal solution of $\bar{\boldsymbol{\mu}}$ is $\bar{\boldsymbol{\mu}}^* = \operatorname{vec}(\bar{\mathbf{M}}^*)$. As we have $\bar{\boldsymbol{\mu}} = \bar{\boldsymbol{\Sigma}} \cdot \bar{\mathbf{z}}$. We construct $f(\bar{\boldsymbol{\mu}})$ as a multi-variates linear objective function. When $\bar{\boldsymbol{\mu}} = \bar{\boldsymbol{\mu}}^*$ is optimal solution, $f(\bar{\boldsymbol{\mu}}) = \mathbf{o}$, where $\mathbf{o}$ is a vector with all zero. Note that

$$
f(\bar{\boldsymbol{\mu}}) = \bar{\boldsymbol{\Sigma}}^{-1} \bar{\boldsymbol{\mu}} - \bar{\mathbf{z}} = \sum_k^K \operatorname{vec}(\mathbf{B}_k \bar{\mathbf{M}} \mathbf{A}_k) - \bar{\mathbf{z}} = \operatorname{vec}(\mathbb{E}[\mathbf{B} \bar{\mathbf{M}} \mathbf{A}]) - \bar{\mathbf{z}}.
\tag{13}
$$

To obtain the optimal solution, we minimize the following problem to obtain an approximate solution $\bar{\mathbf{M}}^*$ of $\bar{\mathbf{M}}$:

$$
\bar{\mathbf{M}}^* = \min_{\bar{\mathbf{M}}} \frac{1}{2} \left\| \sum_k^K \operatorname{vec}(\mathbf{B}_k \bar{\mathbf{M}} \mathbf{A}_k) - \bar{\mathbf{z}} \right\|_2^2.
\tag{14}
$$

The above equation is a quadratic objective, and it can be solved by modern optimization tools efficiently and conveniently.

Since the main objective of the above problem is both convex and Lipschitz smooth w.r.t $\operatorname{vec}(\bar{\mathbf{M}})$, we can use the gradient descent method to solve it with a linear convergence rate (See detailed proof in Appendix F). Here, we use automatic differentiation to calculate the gradient w.r.t. $\bar{\mathbf{M}}$. The overall pseudo Algorithm 1 is in the Appendix A. The transmitted data between the clients and the server is solely $\mathbf{A}_k, \mathbf{B}_k, \mathbf{M}_k$ without any extra auxiliary information, which preserves the data/label privacy for the local clients.

## 3.6 T-SNE OBSERVATION AND DISCUSSIONS

To quickly demonstrate the effectiveness of FedLPA, We show the t-SNE visualization of our FedLPA global model on MNIST dataset with biased training data setting among 10 local clients as an example. The experiment details, t-SNE visualizations of the local models and the global models of other algorithms and discussions are in the Appendix G.2. As shown in Figure 1, FedLPA generates the global model which can distinguish the ten classes, meanwhile, the classes are separate.

# 4 EXPERIMENTS

## 4.1 EXPERIMENTS SETTINGS

**Datasets.** We conduct experiments on MNIST (LeCun et al., 1998), Fashion-MNIST (Xiao et al., 2017), CIFAR-10 (Krizhevsky et al., 2009), and SVHN (Netzer et al., 2011) datasets. We use the data partitioning methods for non-IID settings of the benchmark [1] to simulate different label skews. Specifically, we try two different kinds of partition: 1) #C = $k$: each client only has data from $k$ classes. We first assign $k$ random class IDs for each client. Next, we randomly and equally divide samples of each class to their assigned clients; 2) $p_k$ - Dir($\beta$): for each class, we sample from Dirichlet distribution $p_k$ - Dir($\beta$) and distribute $p_{k,j}$ portion of class $k$ samples to client $j$. In this case, smaller $\beta$ denotes worse skews.

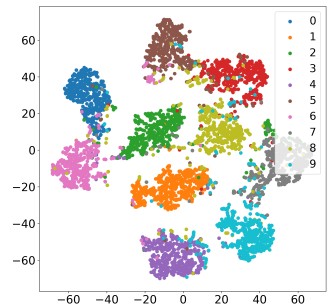

Figure 1: t-SNE visualization for our FedLPA global model.

**Training Details.** By default, we follow Fedavg (McMahan et al., 2017) and other existing studies (Wang et al., 2020c; Li et al., 2022; Diao et al., 2023) to use a simple CNN with 5 layers in our experiments. We set the batch size to 64, the learning rate to 0.001, and the $\lambda = 0.001$ for FedLPA. By default, we set 10 clients and run 200 local epochs for each client. For the various settings of the number of clients and local epochs, we refer to Section 4.3 and Section 4.4. For results with error bars, we run three experiments with different random seeds. All methods were evaluated under fair comparison settings. Due to the page limit, we only present some representative results in the main paper. For more experimental details and results, please refer to Appendix G.

**Baselines.** To ensure fair comparisons, we neglect the comparison with methods that require to download auxiliary models or datasets, such as FedBE (Chen & Chao, 2020), FedKT (Li et al., 2020a) and FedGen (Zhu et al., 2021), or even pretrained large model, like FedDISC (Yang et al., 2023). FedOV (Diao et al., 2023) and FEDCAVE (Heinbaugh et al., 2022) entail sharing more client-side label information or transmitting client label information to the server, which could jeopardize label privacy and are beyond the scope of this study. XorMixFL (Shin et al., 2020) may be not practical as we mentioned before. FedFisher (Jhunjhunwala et al., 2023) is not publicly available. FedDF (Lin et al., 2020), DAFL (Chen et al., 2019) and ADI (Yin et al., 2020) are compared with the state-of-the-art data-free method DENSE (Zhang et al., 2022a). In conclusion, we include one-shot FL algorithms as baselines including Fedavg (McMahan et al., 2017), Fedprox (Li et al., 2020c), Fednova (Wang et al., 2020b), SCAFFOLD (Karimireddy et al., 2020) and the state-of-the-art data-free method DENSE (Zhang et al., 2022a). All the methods are fairly compared, and our implementation is available and the experiment details can be viewed in Appendix G.10.

## 4.2 AN OVERALL COMPARISON

We compare the accuracy between FedLPA and the other baselines as shown in Table 1, the data in the green shadow shows the best results. FedLPA can achieve the best performance in all the dataset and partition settings. In extreme cases such as $\beta = \{0.01, 0.05\}$, #C = 1, #C = 2, FedLPA exhibits a significant performance advantage over the baseline algorithms. This demonstrates our framework's ability to effectively aggregate valuable information from local clients for global weight training. In summary, the state-of-the-art DENSE could be comparable with FedLPA when the skew level is small. However, with the increment of skewness, FedLPA shows significantly superior results.

## 4.3 SCALABILITY

We assess the scalability of FedLPA by varying the number of clients. In this section, we show results on FMNIST in Table 2. From the table, we can observe that FedLPA still almost always achieves the best accuracy when increasing the number of clients. Notably, there is a slight exception highlighted in red, where DENSE outperforms us when we have 20 clients and $\beta = 0.5$, this may

---

[1]https://github.com/Xtra-Computing/NIID-Bench

Table 1: Comparison with various FL algorithms in one round.

| Dataset | Partition | FedLPA | Fednova | SCAFFOLD | Fedavg | Fedprox | DENSE |
|---|---|---|---|---|---|---|---|
| FMNIST | $\beta$=0.01 | 21.20±0.67 | 10.13±0.00 | 15.97±0.12 | 18.17±0.15 | 13.37±0.19 | 15.23±0.14 |
| | $\beta$=0.05 | 54.27±0.38 | 18.67±0.41 | 18.67±0.41 | 18.67±0.41 | 22.03±0.14 | 47.77±0.20 |
| | $\beta$=0.1 | 55.33±0.06 | 30.47±0.59 | 31.40±0.25 | 30.93±0.58 | 31.00±0.52 | 52.93±0.67 |
| | $\beta$=0.3 | 68.20±0.04 | 49.40±0.26 | 46.00±0.02 | 45.17±0.05 | 44.30±0.08 | 64.27±0.08 |
| | $\beta$=0.5 | 73.33±0.06 | 57.03±0.28 | 56.03±0.28 | 59.10±0.63 | 58.10±0.47 | 72.87±0.13 |
| | $\beta$=1.0 | 76.03±0.05 | 63.63±0.33 | 66.10±0.02 | 62.13±0.43 | 63.10±0.29 | 72.97±0.01 |
| | #C=1 | 13.20±0.02 | 10.37±0.00 | 10.40±0.00 | 10.37±0.00 | 13.03±0.18 | 10.00±0.00 |
| | #C=2 | 46.13±0.15 | 21.00±0.10 | 23.53±0.22 | 23.20±0.08 | 19.97±0.10 | 38.90±0.45 |
| | #C=3 | 57.90±0.06 | 27.47±0.02 | 27.37±0.36 | 29.20±0.03 | 23.93±0.33 | 53.40±0.07 |
| CIFAR-10 | $\beta$=0.01 | 16.17±0.00 | 11.57±0.02 | 11.47±0.01 | 11.53±0.05 | 10.47±0.00 | 12.30±0.03 |
| | $\beta$=0.05 | 18.37±0.00 | 10.30±0.00 | 10.73±0.01 | 10.23±0.00 | 10.97±0.02 | 17.87±0.31 |
| | $\beta$=0.1 | 19.97±0.02 | 12.30±0.04 | 10.87±0.01 | 12.83±0.06 | 11.97±0.04 | 19.93±0.07 |
| | $\beta$=0.3 | 26.60±0.01 | 11.77±0.02 | 10.93±0.01 | 10.53±0.00 | 10.97±0.00 | 25.57±0.84 |
| | $\beta$=0.5 | 24.20±0.02 | 11.07±0.00 | 11.77±0.02 | 10.97±0.00 | 11.33±0.00 | 20.17±0.73 |
| | $\beta$=1.0 | 29.33±0.00 | 12.00±0.00 | 13.00±0.00 | 13.23±0.00 | 13.63±0.01 | 28.23±0.34 |
| | #C=1 | 10.70±0.01 | 10.50±0.00 | 10.27±0.00 | 10.23±0.00 | 10.37±0.01 | 10.00±0.00 |
| | #C=2 | 16.40±0.00 | 10.07±0.00 | 12.03±0.08 | 10.07±0.00 | 10.03±0.00 | 14.13±0.22 |
| | #C=3 | 18.97±0.01 | 11.30±0.01 | 11.00±0.01 | 11.53±0.01 | 10.77±0.00 | 14.77±0.11 |
| MNIST | $\beta$=0.01 | 39.17±1.16 | 13.53±0.20 | 8.87±0.01 | 9.37±0.00 | 9.33±0.00 | 15.80±0.24 |
| | $\beta$=0.05 | 70.07±0.05 | 31.60±0.71 | 41.07±0.46 | 38.57±0.28 | 32.23±0.18 | 57.83±1.55 |
| | $\beta$=0.1 | 77.43±0.14 | 48.07±0.28 | 47.73±0.22 | 48.63±0.15 | 47.40±0.00 | 70.33±0.02 |
| | $\beta$=0.3 | 85.77±0.02 | 67.6±0.40 | 67.07±0.15 | 66.17±0.21 | 63.40±0.41 | 84.50±0.01 |
| | $\beta$=0.5 | 88.73±0.07 | 79.27±0.08 | 78.57±0.29 | 77.37±0.07 | 79.60±0.24 | 86.33±0.36 |
| | $\beta$=1.0 | 93.37±0.08 | 84.93±0.18 | 85.33±0.15 | 85.10±0.13 | 86.50±0.16 | 91.43±0.02 |
| | #C=1 | 11.43±0.01 | 10.27±0.02 | 10.10±0.01 | 10.10±0.01 | 10.13±0.01 | 9.93±0.00 |
| | #C=2 | 69.63±0.29 | 20.90±0.49 | 25.23±1.08 | 16.47±0.23 | 14.30±0.34 | 52.73±0.46 |
| | #C=3 | 77.13±0.24 | 29.53±1.65 | 31.83±2.45 | 33.13±2.60 | 29.00±2.05 | 58.90±0.31 |
| SVHN | $\beta$=0.01 | 19.20±0.00 | 13.73±0.14 | 9.83±0.00 | 12.13±0.04 | 11.43±0.12 | 17.33±0.28 |
| | $\beta$=0.05 | 22.93±0.38 | 14.90±0.43 | 15.77±0.14 | 16.60±0.23 | 15.90±0.12 | 21.47±0.20 |
| | $\beta$=0.1 | 39.77±0.69 | 25.97±0.13 | 25.70±0.08 | 22.17±0.02 | 24.50±0.06 | 19.43±0.45 |
| | $\beta$=0.3 | 52.23±0.26 | 34.40±0.28 | 34.03±0.06 | 33.93±0.26 | 34.70±0.20 | 47.13+7.14 |
| | $\beta$=0.5 | 54.27±0.02 | 38.53±0.07 | 40.07±0.13 | 38.53±0.15 | 36.93±0.09 | 53.70±0.07 |
| | $\beta$=1.0 | 67.80±0.01 | 55.60±0.08 | 54.03±0.14 | 55.97±0.04 | 55.23±0.12 | 54.40+9.43 |
| | #C=1 | 19.60±0.00 | 10.43±0.00 | 13.73±0.18 | 13.77±0.17 | 18.27±0.03 | 7.70±0.03 |
| | #C=2 | 47.03±4.63 | 12.90±0.27 | 24.47±0.08 | 20.17±0.04 | 17.47±0.13 | 37.67±0.76 |
| | #C=3 | 48.00±0.22 | 20.87±0.12 | 28.37±0.09 | 27.60±0.03 | 24.93±0.10 | 47.43±0.40 |

Table 2: Experimental results of varying number of clients on FMNIST dataset.

| # of Clients | Partition | FedLPA | Fednova | SCAFFOLD | Fedavg | Fedprox | DENSE |
|---|---|---|---|---|---|---|---|
| 20 Clients | $\beta$=0.01 | 33.57±0.38 | 10.00±0.00 | 13.13±0.24 | 13.23±0.21 | 13.93±0.08 | 10.30±0.00 |
| | $\beta$=0.05 | 47.30±0.74 | 21.30±0.08 | 20.53±0.56 | 21.20±0.64 | 19.40±0.46 | 46.13±0.36 |
| | $\beta$=0.1 | 57.37±0.05 | 31.50±0.29 | 29.23±0.60 | 32.43±0.99 | 28.80±1.26 | 57.20±0.12 |
| | $\beta$=0.3 | 71.30±0.03 | 53.87±0.33 | 50.63±0.10 | 52.83±0.08 | 52.13±0.40 | 71.17±0.04 |
| | $\beta$=0.5 | 74.07±0.00 | 62.83±0.03 | 58.60±0.08 | 60.17±0.03 | 59.47±0.06 | 74.10±0.04 |
| | $\beta$=1.0 | 76.07±0.01 | 68.63±0.08 | 69.13±0.12 | 68.33±0.08 | 69.33±0.10 | 75.47±0.04 |
| | #C=1 | 21.50±0.30 | 10.00±0.00 | 10.00±0.00 | 10.00±0.00 | 10.33±0.00 | 10.00±0.00 |
| | #C=2 | 59.17±0.45 | 19.23±0.23 | 19.47±0.49 | 18.53±0.46 | 13.53±0.26 | 33.07±0.27 |
| | #C=3 | 66.37±0.01 | 27.30±0.20 | 28.07±0.35 | 25.93±0.27 | 24.63±0.26 | 52.23±0.79 |
| 50 Clients | $\beta = 0.01$ | 15.91±0.01 | 10.00±0.00 | 10.00±0.00 | 10.00±0.00 | 10.27±0.00 | 10.00±0.00 |
| | $\beta$=0.05 | 28.43±0.80 | 15.50±0.43 | 17.77±0.25 | 17.37±0.24 | 18.10±0.01 | 25.03±0.47 |
| | $\beta$=0.1 | 57.03±0.00 | 34.33±0.04 | 30.17±0.03 | 28.90±0.05 | 31.00±0.27 | 55.83±0.49 |
| | $\beta$=0.3 | 66.70±0.23 | 46.70±0.12 | 43.97±0.02 | 45.40±0.12 | 45.07±0.11 | 59.23±1.90 |
| | $\beta$=0.5 | 71.13±0.00 | 57.93±0.40 | 52.93±0.22 | 53.67±0.26 | 53.80±0.20 | 69.57±0.02 |
| | $\beta$=1.0 | 71.07±0.04 | 60.00±0.20 | 57.67±0.22 | 56.30±0.45 | 56.90±0.41 | 70.33±0.03 |
| | #C=1 | 15.93±0.02 | 10.00±0.00 | 10.00±0.00 | 10.00±0.00 | 10.27±0.00 | 10.00±0.00 |
| | #C=2 | 49.60±0.37 | 18.03±0.11 | 17.20±0.00 | 20.50±0.26 | 15.70±0.03 | 44.57±0.92 |
| | #C=3 | 65.50±0.05 | 38.03±0.99 | 40.53±1.41 | 40.97±1.51 | 38.93±1.34 | 56.10±0.38 |

be attributed to the dataset being less biased and the DENSE only getting a marginal 0.03% higher test accuracy. Our method is generally much more robust in all kinds of settings.

## 4.4 ABLATION STUDY

The hyper-parameter of our approach is $\lambda$ from Eq. 30, which controls variances of a priori normal distribution and guarantees $\mathbf{A}_k$ and $\mathbf{B}_k$ are positive semi-definite. In this part, we show results on FMNIST. All other Laplace Approximations are sensitive to the hyper-parameter $\lambda$ based on their experimental results, Table 3 shows that our approach is relatively robust. Based on our numerical results, we set $\lambda = 0.001$ by default for our method FedLPA.

We also conduct the experiments when the local epochs are 10,20,50,100. More experiments are available in Appendix G.3, which shows that our methods outperform all the baselines in all kinds of scenarios without requiring extensive tuning.

## 4.5 COMMUNICATION AND COMPUTATION OVERHEAD

Table 3: Experimental results of different hyper-parameter $\lambda$ on FMNIST dataset.

| value of $\lambda$ | 0.01 | 0.001 | 0.0001 |
|---|---|---|---|
| $\beta$=0.01 | 18.63±0.78 | 21.20±0.67 | 22.50±1.84 |
| $\beta$=0.05 | 54.33±0.54 | 54.27±0.38 | 53.30±0.01 |
| $\beta$=0.1 | 56.83±0.19 | 55.33±0.06 | 54.60±0.15 |
| $\beta$=0.3 | 66.83±0.02 | 68.20±0.04 | 67.53±0.03 |
| $\beta$=0.5 | 73.20±0.03 | 73.33±0.06 | 72.17±0.04 |
| $\beta$=1.0 | 76.53±0.02 | 76.03±0.05 | 73.47±0.19 |
| #C=1 | 12.73±0.01 | 13.20±0.02 | 14.17±0.02 |
| #C=2 | 45.20±0.21 | 46.13±0.15 | 44.80±0.03 |
| #C=3 | 58.97±0.07 | 57.90±0.06 | 55.60±0.06 |

Table 4: Communication and computation overhead evaluation.

| | Overall Computation (mins) | Overall Communication (MB) |
|---|---|---|
| FedLPA | 65 | 2.77 |
| Fednova | 50 | 2.47 |
| SCAFFOLD | 50 | 4.94 |
| Fedavg | 50 | 2.47 |
| Fedprox | 75 | 2.47 |
| DENSE | 400 | 2.47 |

We conduct experiments on CIFAR-10 on a single 2080Ti GPU to estimate the overall communication and computation overhead. We set the number of clients is 10. Table 4 shows the numerical results on FedLPA and other baseline approaches. Details of the overhead evaluation are referred to Appendix G.8 and G.9. Our observations reveal that FedLPA is slightly slower than Fednova, SCAFFOLD, Fedavg, and Fedprox, while much faster than the state-of-art data-free approach DENSE. Note that FedLPA has significantly improved the one-shot learning performance of the above four approaches. Similarly, FedLPA performs moderately incremental communication overhead while outperforming other baseline approaches on learning performance. It's noteworthy that FedLPA strikes a favorable balance between computation and communication overhead, making it the most promising approach for one-shot FL.

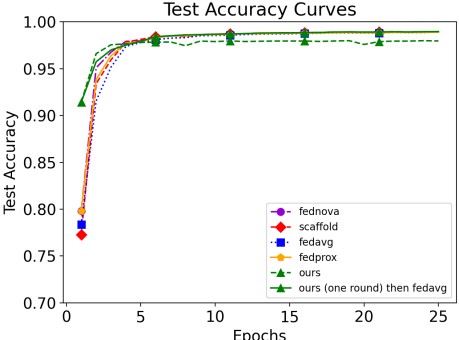

Figure 2: Extension to multiple rounds on MNIST dataset.

### 4.6 SUPPLEMENTARY EXPERIMENTS

**Extension to Multiple Rounds.** We conduct experiments on MNIST with 10 clients and data partitioning $p_k$ - Dir($\beta = 0.5$). The results are shown in Figure 2. As DENSE could not support multiple rounds, we compare our methods with Fedavg, Fednova, SCAFFOLD, and Fedprox. FedLPA achieves the highest accuracy in the first round, denoting the strongest learning capabilities in a one-shot setting. With the increment in the number of rounds, the performances of FedLPA increase slower than the other baseline approaches. This figure shows that the joint approach (ours (one round) then Fedavg) that utilizes FedLPA in the first round and then adopts other baseline methods may be most promising to save communication and computation resources in the multiple-round federated learning scenario.

Some experiments in extreme settings (the number of clients=5, $\beta = 0.001$) and an aggregation visualization can be found in Appendix G.

### 5 CONCLUSIONS

In this work, we design a novel one-shot FL algorithm FedLPA to better model the global parameters in personalized one-shot federated learning. We propose a method that could aggregate the local clients in a layer-wise manner with their posteriors approximation via block-diagonal empirical fisher information matrices, which could effectively capture the accurate statistics of local biased dataset. Our extensive experiments show that FedLPA significantly outperforms other baselines in terms of accuracy under various settings, doing so both effectively and efficiently. Overall, FedLPA stands out as the most practical framework that conducts data-free one-shot FL, particularly well-suited for high data heterogeneity and preserving privacy without extra information leakage.

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

## A    THE FEDLPA ALGORITHM

The proposed algorithm follows the same paradigm as the standard one-shot federated learning framework. The entire procedure is illustrated in Algorithm 1. After completing local training, each client employs its trained local model to calculate the local co-variance over its training dataset using the layer-wise Laplace approximation. Subsequently, each client transmits its local model along with the extra local co-variance information to the server. The server aggregates these contributions to obtain the global expectation, as described in Eq. 7, and trains the global model parameters, as outlined in Eq. 14.

---

**Algorithm 1** FedLPA Algorithm

---

1: **Input:** clients $K$, layers $L$
2: Initialize global weight $\bar{\mathbf{W}}_l$ of layer $l = 1, ..., L$
3: **clients executes:**
4: Initialize local model
5: **for** k = 1, ..., K **do**
6:     $\{\mathbf{M}_{k_l}, \mathbf{A}_{k_l}, \mathbf{B}_{k_l} | l = 1, ..., L\} \leftarrow$ local training
7: **end for**

8: **Server executes:**
9: **for** l = 1, ..., L **do**
10:     $\bar{\mathbf{A}}_l \leftarrow \sum_k^K \mathbf{A}_{k_l}$
11:     $\bar{\mathbf{B}}_l \leftarrow \sum_k^K \mathbf{B}_{k_l}$
12:     $\bar{\mathbf{Z}}_l \leftarrow \sum_k^K \mathbf{B}_{k_l} \mathbf{M}_{k_l} \mathbf{A}_{k_l}$
13:     $\bar{\mathbf{M}}_l \leftarrow$ Train the parameter of the global model
14:     $\bar{\mathbf{W}}_l \leftarrow \bar{\mathbf{M}}_l$
15: **end for**

---

## B    FURTHER DISCUSSION OF EQ. 5

Eq. 5 ignores the first-order term because the first-order derivative is almost zero near the optimal. Terms of a higher order than the second-order terms are ignored because they are much less significant and the curvature near the optima is assumed to be quadratic (Ritter et al., 2018).

In fact, the Gaussian distribution is employed to approximate the small neighborhood of the posteriors based on the MAP estimate. This choice is made because the empirical Fisher information matrix we utilize in the end is equivalent to the expectation of the Hessian of the negative log posterior, as detailed in Appendix C. We are not approximating the entire posterior as Gaussian but rather approximating the local optimization curve near the posterior parameters as Gaussian, where the distance of the points on the curve to the true posterior point is sufficiently small. Thus, it is reasonable to approximate global and local posteriors as multi-variates Gaussian distributions with expectations $\bar{\boldsymbol{\mu}} = \boldsymbol{\theta}^*$ and $\boldsymbol{\mu_k} = \boldsymbol{\theta}_k^*$; co-variances $\bar{\boldsymbol{\Sigma}} = \bar{\mathbf{H}}^{-1}$ and $\boldsymbol{\Sigma}_k = \bar{\mathbf{H}}_k^{-1}$ (Daxberger et al., 2021).

## C    ESTIMATE HESSIAN MATRIX USING EMPIRICAL FISHER INFORMATION MATRIX

We can find that the co-variances are a sum of prior co-variances and the expectations of the second derivation of likelihood.

$$
\begin{aligned}
\boldsymbol{\Sigma}^{-1} &= \bar{\mathbf{H}} \\
&= -\nabla^2 \log p(\boldsymbol{\theta}|\mathcal{D}) \\
&= -\{\nabla^2 \log p(\mathcal{D}|\boldsymbol{\theta}) + \nabla^2 \log p(\boldsymbol{\theta}) - \nabla^2 p(\mathcal{D})\} \\
&= -\mathbb{E}_{s \in \mathcal{D}} \left[\nabla^2 \log p(s|\boldsymbol{\theta})\right] + \lambda \mathbf{I} - 0 \\
&= -\frac{1}{|\mathcal{D}|} \sum_{i=1}^{|\mathcal{D}|} \nabla^2 \log p(s_i|\boldsymbol{\theta}) + \lambda \mathbf{I}.
\end{aligned} \tag{15}
$$

However, $\bar{\mathbf{H}}$ is intractable to compute and store due to many parameters in modern neural networks. An efficient method is to approximate $\bar{\mathbf{H}}$ as a diagonal matrix using the empirical Fisher information matrix $\tilde{\mathbf{F}}$ (Van Loan, 2000).

For a neural network, a $\tilde{\mathbf{F}}$ is defined as below:

$$\tilde{\mathbf{F}} = \mathbb{E}_{p(s|\boldsymbol{\theta})} \left[ \nabla \log p(s|\boldsymbol{\theta}) \nabla \log p(s|\boldsymbol{\theta})^\top \right], \tag{16}$$

where $p(s|\boldsymbol{\theta})$ is the likelihood on data point $x$.

The $\tilde{\mathbf{F}}$ has an obvious property that it is equivalent to the expectation of the Hessian of the negative log posterior, as shown below:

$$\begin{aligned}
\bar{\mathbf{H}} &= \mathbb{E}_{p(s|\boldsymbol{\theta})} \left[ \mathbf{H}_{-\log p(s|\boldsymbol{\theta})} \right] \\
&= -\mathbb{E}_{p(s|\boldsymbol{\theta})} \left[ \mathbf{H}_{\log p(s|\boldsymbol{\theta})} \right] \\
&= -\mathbb{E}_{p(s|\boldsymbol{\theta})} \left[ \mathbf{J} \left( \frac{\nabla p(s|\boldsymbol{\theta})}{p(s|\boldsymbol{\theta})} \right) \right] \\
&= -\mathbb{E}_{p(s|\boldsymbol{\theta})} \left[ \frac{\mathbf{H}_{p(s|\boldsymbol{\theta})} p(s|\boldsymbol{\theta}) - \nabla p(s|\boldsymbol{\theta}) \nabla p(s|\boldsymbol{\theta})^\top}{p(s|\boldsymbol{\theta}) p(s|\boldsymbol{\theta})} \right] \\
&= -\mathbb{E}_{p(s|\boldsymbol{\theta})} \left[ \frac{\mathbf{H}_{p(s|\boldsymbol{\theta})} p(s|\boldsymbol{\theta})}{p(s|\boldsymbol{\theta}) p(s|\boldsymbol{\theta})} - \frac{\nabla p(s|\boldsymbol{\theta}) \nabla p(s|\boldsymbol{\theta})^\top}{p(s|\boldsymbol{\theta}) p(s|\boldsymbol{\theta})} \right] \\
&= -\mathbb{E}_{p(s|\boldsymbol{\theta})} \left[ \frac{\mathbf{H}_{p(s|\boldsymbol{\theta})}}{p(s|\boldsymbol{\theta})} - \left( \frac{\nabla p(s|\boldsymbol{\theta})}{p(s|\boldsymbol{\theta})} \right) \left( \frac{\nabla p(s|\boldsymbol{\theta})}{p(s|\boldsymbol{\theta})} \right)^\top \right] \\
&= -\mathbb{E}_{p(s|\boldsymbol{\theta})} \left[ \frac{\mathbf{H}_{p(s|\boldsymbol{\theta})}}{p(s|\boldsymbol{\theta})} \right] + \mathbb{E}_{p(s|\boldsymbol{\theta})} \left[ \left( \frac{\nabla p(s|\boldsymbol{\theta})}{p(s|\boldsymbol{\theta})} \right) \left( \frac{\nabla p(s|\boldsymbol{\theta})}{p(s|\boldsymbol{\theta})} \right)^\top \right] \\
&= \mathbb{E}_{p(s|\boldsymbol{\theta})} \left[ \nabla \log p(s|\boldsymbol{\theta}) \nabla \log p(s|\boldsymbol{\theta})^\top \right] - \int \frac{\mathbf{H}_{p(s|\boldsymbol{\theta})}}{p(s|\boldsymbol{\theta})} p(s|\boldsymbol{\theta}) \mathrm{d}x \\
&= \tilde{\mathbf{F}} - \mathbf{H}_{\int p(s|\boldsymbol{\theta}) \mathrm{d}x} \\
&= \tilde{\mathbf{F}} - \mathbf{H}_1 \\
&= \tilde{\mathbf{F}}.
\end{aligned} \tag{17}$$

However, the expectation of the likelihood $p(s|\boldsymbol{\theta})$ is usually intractable to compute. Therefore, a common method is to use an empirical distribution $q(s) \approx p(s|\boldsymbol{\theta})$ to approximate the likelihood, and $q(s)$ is always assumed to be $p(\mathcal{D})$.

$$\begin{aligned}
\tilde{\mathbf{F}} &\approx \mathbb{E}_{q(s)} \left[ \nabla \log p(s|\boldsymbol{\theta}) \nabla \log p(s|\boldsymbol{\theta})^\top \right] \\
&= \mathbb{E} \left[ \nabla \log p(s|\boldsymbol{\theta}) \nabla \log p(s|\boldsymbol{\theta})^\top \right] \\
&= \frac{1}{|\mathcal{D}|} \sum_{i=1}^{|\mathcal{D}|} \nabla \log p(s_i|\boldsymbol{\theta}) \nabla \log p(s_i|\boldsymbol{\theta})^\top.
\end{aligned} \tag{18}$$

Eq. 18 is called the empirical Fisher information matrix, which is computable. Therefore, a diagonal matrix of $\bar{\mathbf{H}}$ can be estimated by the diagonal of empirical $\tilde{\mathbf{F}}$. We denote $\mathrm{diag}(\cdot)$ as the diagonal of a matrix.

$$\begin{aligned}
\mathrm{diag}(\bar{\mathbf{H}}) &= \mathrm{diag}(\tilde{\mathbf{F}} + \lambda \mathbf{I}) \\
&\approx \frac{1}{|\mathcal{D}|} \sum_{i=1}^{|\mathcal{D}|} \mathrm{diag}(\nabla \log p(x|\boldsymbol{\theta}) \nabla \log p(x|\boldsymbol{\theta})^\top) + \lambda \mathbf{I}.
\end{aligned} \tag{19}$$

## D  Block-Diagonal Empirical Fisher Information Matrix

Storing and computing Eq. 19 for a neural network is efficient. It is equivalent to calculating the expectation of the square of the gradient of the loss function with respect to the parameters $\boldsymbol{\theta}$. Meanwhile, $\mathrm{diag}(\bar{\mathbf{H}})$ is easy to inverse, $\mathrm{diag}(\bar{\mathbf{H}})^{-1} = \frac{1}{\mathrm{diag}(\bar{\mathbf{H}})}$. Therefore, such a diagonal matrix is always used as the co-variances of a Gaussian posterior, that is:

$$p(\boldsymbol{\theta}|\mathcal{D}) \equiv \boldsymbol{\theta} \sim \mathcal{N}(\boldsymbol{\theta}^*, \mathrm{diag}(\tilde{\mathbf{F}} + \lambda\mathbf{I})^{-1}). \tag{20}$$

Separately, as for layers $l$, the weight $\mathrm{vec}(\mathbf{W}_l)$ is approximated by a Gaussian,

$$\mathrm{vec}(\mathbf{W}_l) \sim \mathcal{N}(\mathrm{vec}(\mathbf{W}_l^*), \mathrm{diag}(\tilde{\mathbf{F}}_l + \lambda\mathbf{I})^{-1}). \tag{21}$$

This diagonal approximation has been used in many previous works (Martens, 2016; Martens et al., 2010; Liu et al., 2021; Kirkpatrick et al., 2017), but it is crude that ignores co-relations between different parameters and only considers the self-relations of parameters. As a result, regions with high posterior probability due to strong correlation are incorrectly discarded.

To efficiently capture these correlations between different parameters, earlier studies (Martens & Grosse, 2015a; Grosse & Martens, 2016; Botev et al., 2017) have proposed estimating a block-diagonal Fisher information matrix $\mathbf{F}$ instead of a coordinate-wise one.

In a $L$ layers neural network, there are $L \times L$ blocks in its empirical Fisher. For layers $l_1$ and $l_2$, their empirical Fisher block is as below:

$$\tilde{\mathbf{F}}_{l_1,l_2} = \mathbb{E}\left[\nabla \log p(s|\mathrm{vec}(\mathbf{W}_{l_1}))\nabla \log p(s|\mathrm{vec}(\mathbf{W}_{l_2})^\top\right], \tag{22}$$

In modern back-propagation neural networks, the weights are usually updated by the back-propagation algorithm. We denote $\tilde{y} = f(x, \boldsymbol{\theta})$ as the output of the network $f(\cdot, \boldsymbol{\theta})$; $\mathcal{L}(\cdot)$ as the loss function. Given a sample $s = (x, y)$, the loss is $\mathcal{L}(y, \tilde{y})$.

After computing the derivation of loss function $g_{\mathrm{loss}} = \frac{\partial \mathcal{L}(y, f(x, \boldsymbol{\theta}))}{\partial \tilde{y}}$, the gradients used to update weights are calculated by back-propagation algorithm.

Therefore, for layers $l$, the gradients are as below,

$$\begin{aligned}
\mathbf{G}_l &= -\nabla \log p(s|\mathbf{W}_l) \\
&= \frac{\partial \mathcal{L}(y, f(x, \boldsymbol{\theta}))}{\partial \mathrm{vec}(\mathbf{W}_l)} \\
&= \hat{\mathbf{b}}_l \hat{\mathbf{a}}_{l-1}^\top,
\end{aligned} \tag{23}$$

where $\hat{\mathbf{b}}_l = \mathbf{W}_{l+1}^\top \hat{\mathbf{b}}_{l+1} \odot \sigma_l'(\mathbf{h}_l)$ and $\hat{\mathbf{b}}_L = \mathbf{W}_L^\top g_{\mathrm{loss}} \odot \sigma_L'(\mathbf{h}_{L-1})$. $\odot$ is coordinate-wise multiplication. All intermediate values in Eq. 23 can be computed efficiently by back-propagation algorithm (Rumelhart et al., 1986; Martens & Grosse, 2015a).

Applying Eq. 23 into Eq. 22, we get a simply format of $\tilde{\mathbf{F}}_{l_1,l_2}$ represented by $\mathbf{G}_{l_1}$ and $\mathbf{G}_{l_2}$,

$$\tilde{\mathbf{F}}_{l_1,l_2} = \mathbb{E}\left[\mathrm{vec}(\mathbf{G}_{l_1})\mathrm{vec}(\mathbf{G}_{l_2})^\top\right]. \tag{24}$$

Using Kronecker product property, we re-write $\mathrm{vec}(\mathbf{G}_l)$ as a Kronecker product of $\hat{a}_{l-1}$ and $\hat{b}_l$,

$$\mathrm{vec}(\mathbf{G}_l) = \mathrm{vec}(\hat{\mathbf{b}}_l \hat{\mathbf{a}}_{l-1}^\top) = \hat{\mathbf{a}}_{l-1} \otimes \hat{\mathbf{b}}_l. \tag{25}$$

Therefore, using Eq. 25, we get a new format of $\tilde{\mathbf{F}}_{l_1,l_2}$,

$$\begin{aligned}
\tilde{\mathbf{F}}_{l_1,l_2} &= \mathbb{E}\left[(\hat{\mathbf{a}}_{l_1-1} \otimes \hat{\mathbf{b}}_{l_1})(\hat{\mathbf{a}}_{l_2-1} \otimes \hat{\mathbf{b}}_{l_2})^\top\right] \\
&= \mathbb{E}\left[\hat{\mathbf{a}}_{l_1-1}\hat{\mathbf{a}}_{l_2-1}^\top \otimes \hat{\mathbf{b}}_{l_1}\hat{\mathbf{b}}_{l_2}^\top\right].
\end{aligned} \tag{26}$$

However, until the above step, there is still a significant overhead to store and inverse $\tilde{\mathbf{F}}_{l_1,l_2}$. In order to save computational overhead, previous work (Martens & Grosse, 2015a) takes an approximate matrix $\hat{\mathbf{F}}_{l_1,l_2}$. $\hat{\mathbf{F}}_{l_1,l_2}$ is a result of approximating the expectation of the Kronecker product by the Kronecker product of expectation. The approximation is as below:

$$
\begin{aligned}
\tilde{\mathbf{F}}_{l_1,l_2} &= \mathbb{E}\left[\hat{\mathbf{a}}_{l_1-1}\hat{\mathbf{a}}_{l_2-1}^\top \otimes \hat{\mathbf{b}}_{l_1}\hat{\mathbf{b}}_{l_2}^\top\right] \approx \hat{\mathbf{F}}_{l_1,l_2} \\
\hat{\mathbf{F}}_{l_1,l_2} &= \mathbb{E}\left[\hat{\mathbf{a}}_{l_1-1}\hat{\mathbf{a}}_{l_2-1}^\top\right] \otimes \mathbb{E}\left[\hat{\mathbf{b}}_{l_1}\hat{\mathbf{b}}_{l_2}^\top\right] \\
&= \hat{\mathbf{A}}_{l_1-1,l_2-1} \otimes \hat{\mathbf{B}}_{l_1,l_2},
\end{aligned}
\tag{27}
$$

where $\hat{\mathbf{A}}_{l_1-1,l_2-1} = \hat{\mathbf{a}}_{l_1-1}\hat{\mathbf{a}}_{l_2-1}^\top$ and $\hat{\mathbf{B}}_{l_1,l_2} = \hat{\mathbf{b}}_{l_1}\hat{\mathbf{b}}_{l_2}^\top$.

It is important to note that, however, the expectation of a Kronecker product is not equal to the Kronecker product of expectations. Such an approximation is equivalent to assuming that $\hat{\mathbf{a}}_{l-1}$ and $\hat{\mathbf{b}}_l$ are independent, but it is obviously not true because they are weakly related due to back-propagation. Although the approximation is biased, it could well capture the coarse structure of the empirical Fisher information matrix as demonstrated in Martens & Grosse (2015a) and fit in the personalized one-shot federated learning settings to obtain the information of vital importance, which may also explain the performance result for the multiple round setting in Figure 2. The visualization of the block-diagonal Fisher information matrix is shown in Appendix D.1. Meanwhile, Eq. 27 saves significant computational overhead of inverting Kronecker Product and small sizes of $\hat{\mathbf{A}}_{l_1-1,l_2-1}$ and $\hat{\mathbf{B}}_{l_1,l_2}$.

By filling all blocks into $\tilde{\mathbf{F}}$, we obtain $\tilde{\mathbf{F}}$ in a format of the Khatri-Rao product,

$$
\tilde{\mathbf{F}} \approx \hat{\mathbf{F}} = \begin{bmatrix}
\hat{\mathbf{A}}_{0,0} \otimes \hat{\mathbf{B}}_{1,1} & \hat{\mathbf{A}}_{0,1} \otimes \hat{\mathbf{B}}_{1,2} & \cdots & \hat{\mathbf{A}}_{0,L-1} \otimes \hat{\mathbf{B}}_{1,L} \\
\hat{\mathbf{A}}_{1,0} \otimes \hat{\mathbf{B}}_{2,1} & \hat{\mathbf{A}}_{1,1} \otimes \hat{\mathbf{B}}_{2,2} & \cdots & \hat{\mathbf{A}}_{1,L-1} \otimes \hat{\mathbf{B}}_{2,L} \\
\vdots & \vdots & \ddots & \vdots \\
\hat{\mathbf{A}}_{L-1,0} \otimes \hat{\mathbf{B}}_{L,1} & \hat{\mathbf{A}}_{L-1,1} \otimes \hat{\mathbf{B}}_{\ell,2} & \cdots & \hat{\mathbf{A}}_{L-1,L-1} \otimes \hat{\mathbf{B}}_{L,L}
\end{bmatrix}
\tag{28}
$$

Therefore, the Gaussian posterior is given,

$$
p(\boldsymbol{\theta}|\mathcal{D}) \equiv \boldsymbol{\theta} \sim \mathcal{N}(\boldsymbol{\theta}^*, (\hat{\mathbf{F}} + \lambda\mathbf{I})^{-1}).
\tag{29}
$$

The prior co-variances $\lambda^{-1}\mathbf{I}$ are only added into a diagonal block of $\hat{\mathbf{F}}$, but it is not convenient to be inverted. Similar to an approximation in Eq. 27, we add prior co-variances inside Kronecker product,

$$
\begin{aligned}
\hat{\mathbf{F}}_{l,l} + \lambda\mathbf{I} &= \hat{\mathbf{A}}_{l-1,l-1} \otimes \hat{\mathbf{B}}_{l,l} + \lambda\mathbf{I} \otimes \mathbf{I} \\
&\approx \mathbf{F}_{l,l} \\
&= (\hat{\mathbf{A}}_{l-1,l-1} + \pi_l\sqrt{\lambda}\mathbf{I}) \otimes (\hat{\mathbf{B}}_{l,l} + \frac{1}{\pi_l}\sqrt{\lambda}\mathbf{I}) \\
&= \hat{\mathbf{A}}_{l-1,l-1} \otimes \hat{\mathbf{B}}_{l,l} + \pi_l\sqrt{\lambda}\mathbf{I} \otimes \hat{\mathbf{B}}_{l,l} \\
&\quad + \frac{1}{\pi_l}\sqrt{\lambda}\hat{\mathbf{A}}_{l-1,l-1} \otimes \mathbf{I} + \lambda\mathbf{I} \otimes \mathbf{I},
\end{aligned}
\tag{30}
$$

where $\pi_l$ is a factor minimizing approximation error in $\hat{\mathbf{F}}_{l,l}$. Previous works (Martens & Grosse, 2015a; Grosse & Martens, 2016) give it formula as below,

$$
\pi_l = \sqrt{\frac{\left\|\hat{\mathbf{A}}_{l-1,l-1} \otimes \mathbf{I}\right\|}{\left\|\mathbf{I} \otimes \hat{\mathbf{B}}_{l,l}\right\|}}.
\tag{31}
$$

Previous work (Martens & Grosse, 2015a) shows that co-relations inner a layer are much more significant than others. Therefore, it is reasonable to assume that layers are independent of each

other, which means the Fisher information matrix is block-diagonal. Then the posterior for layer $l$ is given in Martens & Grosse (2015a) as below,

$$\text{vec}(\mathbf{W}_l) \sim \mathcal{N}(\text{vec}(\mathbf{W}_l^*), \mathbf{F}_{l,l}^{-1}). \tag{32}$$

Additionally, keeping co-relations between layers is feasible using Eq. 28. Researchers in Martens & Grosse (2015a) have proposed a tri-diagonal approximation for the Fisher information matrix, which considers the co-relations between two consecutive layers, but the effect is not significant compared with the required extra computational resource.

Therefore, in our methodology, we apply the Layer-wise Laplace approximation to obtain a block-diagonal co-variance of the posterior.

For simplicity, for the $k$-th client, we use $\mathbf{A}_{k_l} = \hat{\mathbf{A}}_{l-1,l-1} + \pi_l \sqrt{\lambda} \mathbf{I} = \hat{\mathbf{a}}_{k_{l-1}} \hat{\mathbf{a}}_{k_{l-1}}^\top + \pi_l \sqrt{\lambda} \mathbf{I}$; We use $\mathbf{B}_{k_l} = \hat{\mathbf{B}}_{l,l} + \frac{1}{\pi_l} \sqrt{\lambda} \mathbf{I} = \hat{\mathbf{b}}_{k_l} \hat{\mathbf{b}}_{k_l}^\top + \frac{1}{\pi_l} \sqrt{\lambda} \mathbf{I}$

For layer $l$ on client $k$, its empirical Fisher block is as below,

$$\boldsymbol{\Sigma}_{k_l}^{-1} \approx \mathbf{F}_{k_l} = \mathbf{A}_{k_l} \otimes \mathbf{B}_{k_l}, \tag{33}$$

## D.1 Visualization of the Block-diagonal Empirical Fisher Information Matrix

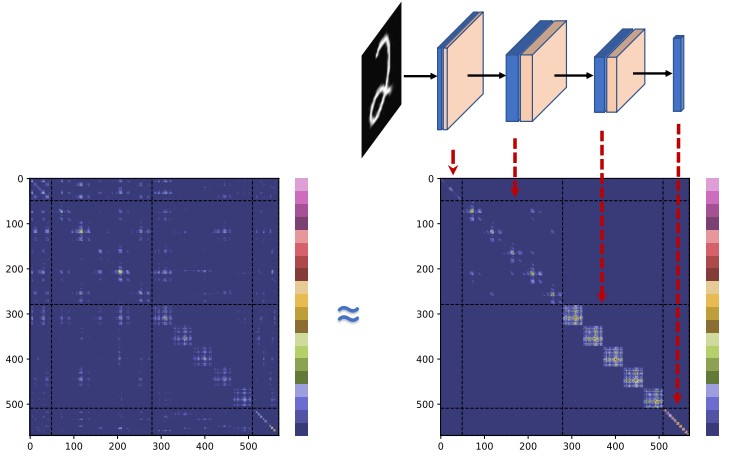

Figure 3: Visualization of block-diagonal empirical Fisher information matrix.

In Figure 3, we give an example, showing that the block-diagonal empirical Fisher information matrix could well capture the coarse structure of the full Fisher information matrix based on a neural network with three convolution layers and one fully connected layer trained on MNIST dataset. The four diagonal blocks corresponding to four layers, can capture the most useful information of the original Fisher Information Matrix. Thus, our proposed FedLPA can yield good performance with this approximation.

## E Expectation Approximation (EA)

Previous works (Martens & Grosse, 2015a; Grosse & Martens, 2016) approximate the expectation of Kronecker products by a Kronecker product of expectations $\mathbb{E}[\mathbf{A} \otimes \mathbf{B}] \approx \mathbb{E}[\mathbf{A}] \otimes \mathbb{E}[\mathbf{B}]$ with an assumption of $\mathbf{A}_{k_l}$ and $\mathbf{B}_{k_l}$ are independent, which is called Expectation Approximation (EA).

It is a simple and effective method to approximate the expectation of Kronecker products. As a result, the global co-variance $\bar{\boldsymbol{\Sigma}}$ is approximated by:

$$\bar{\boldsymbol{\Sigma}}_l \approx (\sum_k^K \mathbf{A}_{k_l})^{-1} \otimes (\sum_k^K \mathbf{B}_{k_l})^{-1} = \bar{\mathbf{A}}_l^{-1} \otimes \bar{\mathbf{B}}_l^{-1}, \tag{34}$$

where $\bar{\mathbf{A}}_l = \sum_k^K \mathbf{A}_{k_l}$ and $\bar{\mathbf{B}}_l = \sum_k^K \mathbf{B}_{k_l}$. Denoting $\bar{\mathbf{Z}}_l$ as matrix formula of $\bar{\mathbf{z}}_l = \text{vec}(\bar{\mathbf{Z}}_l)$, then $\bar{\boldsymbol{\mu}}_l$ can be computed efficiently as below:

$$\bar{\boldsymbol{\mu}}_l = \bar{\boldsymbol{\Sigma}}_l \cdot \bar{\mathbf{z}}_l \approx (\bar{\mathbf{A}}_l^{-1} \otimes \bar{\mathbf{B}}_l^{-1})\bar{\mathbf{z}}_l = \text{vec}(\bar{\mathbf{B}}_l^{-1}\bar{\mathbf{Z}}_l\bar{\mathbf{A}}_l^{-1}). \tag{35}$$

However, Eq. 35 leads to a biased global expectation. The EA needs the assumption of independence, but $\mathbf{A}_{k_l}$ and $\mathbf{B}_{k_l}$ are weakly related in back-propagation. Besides, even if they are independent, Eq. 35 still suffers from approximation error because the clients' number $K$ is finite and always a small number but statistical independence can only be demonstrated when the sampling number is large enough. Eq. 36 shows the approximation error directly:

$$\begin{aligned}(\mathbf{A}_1 + \mathbf{A}_2) \otimes (\mathbf{B}_1 + \mathbf{B}_2) &= \mathbf{A}_1 \otimes \mathbf{B}_1 + \mathbf{A}_2 \otimes \mathbf{B}_2 \\ &\quad + \mathbf{A}_1 \otimes \mathbf{B}_2 + \mathbf{A}_2 \otimes \mathbf{B}_1 \\ &\neq \mathbf{A}_1 \otimes \mathbf{B}_1 + \mathbf{A}_2 \otimes \mathbf{B}_2.\end{aligned} \tag{36}$$

# F    CONVERGENCE ANALYSIS OF EQ. 14

We denote $\mathbf{v} = \text{vec}(\bar{\mathbf{M}})$. For $\mathbf{v} \in \mathcal{V}$, we have $\|\mathbf{v}\|^2 \leq D$ and $D$ is a positive constant. We let $g(\mathbf{v}) = \frac{1}{2}\|\sum_k^K \text{vec}(\mathbf{B}_k\bar{\mathbf{M}}\mathbf{A}_k) - \bar{\mathbf{z}}\|_2^2$. Then we have

$$g(\mathbf{v}) = \frac{1}{2}\|(\sum_k^K \mathbf{A}_k^\top \otimes \mathbf{B}_k)\text{vec}(\bar{\mathbf{M}}) - \bar{\mathbf{z}}\|_2^2 = \frac{1}{2}\|(\sum_k^K \mathbf{A}_k^\top \otimes \mathbf{B}_k)\mathbf{v} - \bar{\mathbf{z}}\|_2^2. \tag{37}$$

Then we have

$$\nabla_\mathbf{v} g(\mathbf{v}) = \mathbf{v}^\top (\sum_k^K \mathbf{A}_k^\top \otimes \mathbf{B}_k)^\top (\sum_k^K \mathbf{A}_k^\top \otimes \mathbf{B}_k) - \bar{\mathbf{z}}^\top (\sum_k^K \mathbf{A}_k^\top \otimes \mathbf{B}_k). \tag{38}$$

Therefore, for any two points $\mathbf{v}_1, \mathbf{v}_2$, we have

$$\|\nabla_\mathbf{v} g(\mathbf{v}_1) - \nabla_\mathbf{v} g(\mathbf{v}_2)\| \leq \|\mathbf{v}_1 - \mathbf{v}_2\|\|(\sum_k^K \mathbf{A}_k^\top \otimes \mathbf{B}_k)^\top (\sum_k^K \mathbf{A}_k^\top \otimes \mathbf{B}_k)\|, \tag{39}$$

where the inequality is due to the Cauchy-Schwarz inequality. Since $\mathbf{A}_k$ and $\mathbf{B}_k$ are all constant, there exists a positive constant $L$ s.t. $\|\nabla_\mathbf{v} g(\mathbf{v}_1) - \nabla_\mathbf{v} g(\mathbf{v}_2)\| \leq L\|\mathbf{v}_1 - \mathbf{v}_2\|\|$. Therefore $g(\mathbf{v})$ is $L$-Lipschitz smooth. Moreover, we have $\nabla_\mathbf{v}^2 g(\mathbf{v}) = (\sum_k^K \mathbf{A}_k^\top \otimes \mathbf{B}_k)^\top (\sum_k^K \mathbf{A}_k^\top \otimes \mathbf{B}_k)$, which is positive semi-definite. So $g(\mathbf{v})$ is convex w.r.t $\mathbf{v}$. Therefore, let $\delta = \frac{1}{L}$, then we have $g(\mathbf{v}^{t+1}) \leq g(\mathbf{v}^t) - \frac{1}{2L}\|\nabla g(\mathbf{v}^t)\|^2$. We have

$$\|\mathbf{v}^{t+1} - \mathbf{v}^*\|^2 = \|\mathbf{v}^t - \delta\nabla g(\mathbf{v}^t) - \mathbf{v}^*\|^2 \tag{40}$$

$$= \|\mathbf{v}^t - \mathbf{v}^*\|^2 - \frac{2}{L}(\langle \nabla g(\mathbf{v}^t)^\top, \mathbf{v}^t - \mathbf{v}^*\rangle - \frac{1}{2L}\|\nabla g(\mathbf{v}^t)\|^2). \tag{41}$$

Then we obtain

$$\sum_{t=0}^{T-1} g(\mathbf{v}^{t+1}) - g(\mathbf{v}^*) \leq \sum_{t=0}^{T-1}(g(\mathbf{v}^t) - g(\mathbf{v}^*) - \frac{1}{2L}\|\nabla g(\mathbf{v}^t)\|^2) \tag{42}$$

$$\leq \sum_{t=0}^{T-1}(\langle \nabla g(\mathbf{v}^t)^\top, \mathbf{v}^t - \mathbf{v}^*\rangle - \frac{1}{2L}\|\nabla g(\mathbf{v}^t)\|^2) \tag{43}$$

$$= \frac{L}{2}\sum_{t=0}^{T-1}(\|\mathbf{v}^t - \mathbf{v}^*\|^2 - \|\mathbf{v}^{t+1} - \mathbf{v}^*\|^2) \tag{44}$$

$$\leq \frac{L}{2}\|\mathbf{v}^0 - \mathbf{v}^*\|^2. \tag{45}$$

Therefore, we obtain $g(\mathbf{v}^{t+1}) - g(\mathbf{v}^*) \leq \frac{L}{2t}\|\mathbf{v}^0 - \mathbf{v}^*\|^2 \leq \frac{LD^2}{2t}$, it has a linear convergent rate. Note that $\mathbf{v} = \text{vec}(\bar{\mathbf{M}})$, this also implies that we have a linear convergent rate to $\bar{\mathbf{M}}^*$.

# G ADDITIONAL EXPERIMENTS

## G.1 DATASET INFORMATION

Our experiments were conducted using four real-world datasets: MNIST (LeCun et al., 1998), FM-NIST (Xiao et al., 2017), SVHN (Netzer et al., 2011), and CIFAR-10 (Krizhevsky et al., 2009). Here's a brief overview of these datasets:

MNIST Dataset: The MNIST dataset comprises binary images of handwritten digits. It consists of 60,000 28x28 training images and 10,000 testing images.

FMNIST Dataset: Similar to MNIST, the FMNIST dataset also contains 60,000 28x28 training images and 10,000 testing images.

SVHN Dataset: The SVHN dataset includes 73,257 32x32 color training images and 10,000 testing images.

CIFAR-10 Dataset: CIFAR-10 consists of 60,000 32x32 color images distributed across ten classes, with each class containing 6,000 images.

The input dimensions for MNIST, FMNIST, SVHN, and CIFAR-10 are 784, 784, 3,072, and 3,072, respectively.

## G.2 T-SNE VISUALIZATION

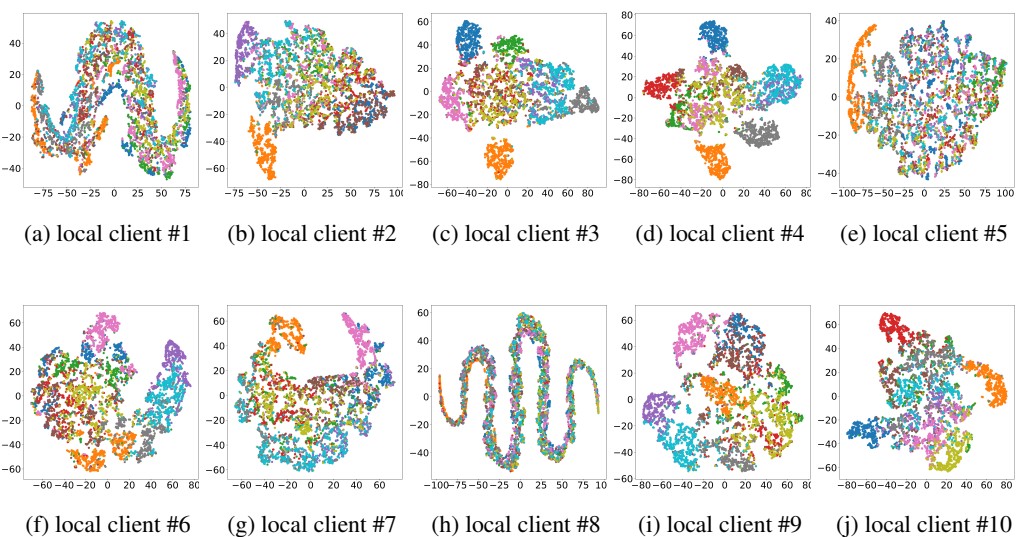

(a) local client #1    (b) local client #2    (c) local client #3    (d) local client #4    (e) local client #5

(f) local client #6    (g) local client #7    (h) local client #8    (i) local client #9    (j) local client #10

Figure 4: t-SNE visualizations of 10 local clients.

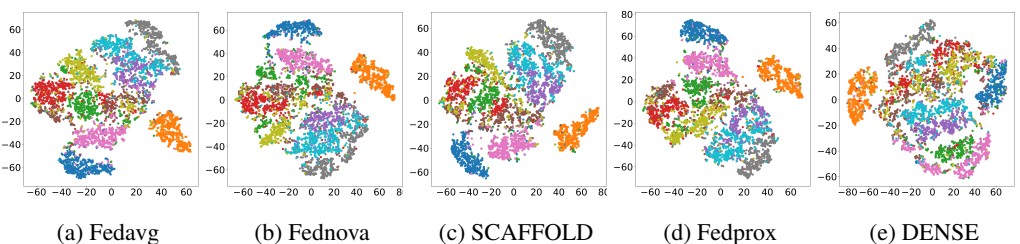

(a) Fedavg    (b) Fednova    (c) SCAFFOLD    (d) Fedprox    (e) DENSE

Figure 5: t-SNE visualizations of the baseline approaches on the global model.

We conduct experiments using MNIST dataset with a $\beta$ value of 0.05, training 10 local clients over 200 local epochs with random seed 0. In this biased local dataset setting, local clients could only distinguish a subset of the classes, as illustrated in Figure 4.

Based on seed 0, we partition the training data for the 10 local clients with the following form (label:# of the data) as:

local client #1: {4: 2, 5: 12, 6: 2847, 9: 16}
local client #2: {1: 20, 4: 189, 5: 5349}
local client #3: {0: 669, 1: 476, 2: 67, 6: 15, 7: 6068}
local client #4: {0: 266, 1: 375, 3: 3956, 7: 196, 9: 5932}
local client #5: {0: 4, 1: 418, 2: 5862}
local client #6: {1: 2, 2: 25, 4: 5195, 5: 24, 6: 80, 8: 28}
local client #7: {1: 5034, 2: 3, 4: 22, 5: 6, 6: 2669}
local client #8: {0: 4914}
local client #9: {4: 433, 5: 29, 6: 307, 8: 5373}
local client #10: {0: 70, 1: 417, 2: 1, 3: 2175, 4: 1, 5: 1, 7: 1, 8: 450, 9: 1}

It is worth noting that local client #2 has the training data mostly with label number 5, and as the corresponding t-SNE visualization shows in Figure 4b, the local train model could mainly cluster the data with label 5 (marked as purple). As data for label 1 (marked as orange) is different from other data with all other labels, some local clients may be able to cluster the data with label 1 with good results. Other local clients, such as local client #3, #4, #6, #7, #9, #10, show the similar results like local client #2.

Figure 5 displays the t-SNE visualization for the global models of Fedavg, Fednova, SCAFFOLD, Fedprox, and DENSE using the training data, with the figure legends identical to those in Figure 1. It's evident from Figure 1 that FedLPA outperforms the baselines in classifying the ten classes. FedLPA's superiority is not only demonstrated by its ability to cluster the ten classes but also by the distinct separation between classes, as observed in Figure 1, compared to the baselines.

## G.3 EXPERIMENTS ON DIFFERENT LOCAL EPOCH NUMBERS

Table 5: Comparison with various FL algorithms in one round with 10 local epochs settings.

| Dataset | Partition | FedLPA | Fednova | SCAFFOLD | Fedavg | Fedprox | DENSE |
|---------|-----------|--------|---------|----------|--------|---------|-------|
| FMNIST | $\beta$=0.01 | 24.47±1.02 | 11.43±0.04 | 13.37±0.11 | 11.83±0.03 | 12.30±0.10 | 10.00±0.00 |
| | $\beta$=0.05 | 30.77±0.94 | 15.67±0.28 | 20.07±0.46 | 19.93±0.30 | 20.07±0.37 | 23.60±0.14 |
| | $\beta$=0.1 | 42.83±0.33 | 25.10±1.32 | 21.03±1.08 | 22.57±1.08 | 22.20±1.17 | 34.83±0.16 |
| | $\beta$=0.3 | 61.43±0.17 | 40.90±0.05 | 40.70±0.09 | 38.20±0.10 | 37.50±0.03 | 43.17±0.05 |
| | $\beta$=0.5 | 67.63±0.36 | 52.43±0.60 | 51.77±0.46 | 54.67±0.73 | 54.33±0.64 | 54.30±0.07 |
| | $\beta$=1.0 | 71.90±0.09 | 51.03±0.62 | 52.30±0.53 | 51.50±0.62 | 50.90±0.57 | 52.30±0.15 |
| | #C=1 | 13.03±0.04 | 10.90±0.02 | 11.27±0.03 | 10.90±0.02 | 11.43±0.04 | 10.00±0.00 |
| | #C=2 | 28.93±0.74 | 16.93±0.19 | 22.07±0.01 | 23.83±0.20 | 23.33±0.03 | 22.60±0.88 |
| | #C=3 | 37.73±0.09 | 22.20±0.23 | 26.60±0.03 | 23.67±0.27 | 22.80±0.40 | 23.03±0.86 |
| CIFAR-10 | $\beta$=0.01 | 15.80±0.00 | 10.07±0.00 | 12.13±0.09 | 11.90±0.07 | 11.93±0.07 | 10.00±0.00 |
| | $\beta$=0.05 | 20.23±0.01 | 10.90±0.02 | 10.00±0.00 | 10.00±0.00 | 10.00±0.00 | 13.33±0.04 |
| | $\beta$=0.1 | 20.20±0.07 | 10.27±0.00 | 10.93±0.02 | 10.37±0.00 | 10.27±0.00 | 14.77±0.09 |
| | $\beta$=0.3 | 25.60±0.01 | 18.13±0.33 | 14.97±0.05 | 14.77±0.05 | 15.67±0.03 | 20.33±0.06 |
| | $\beta$=0.5 | 25.60±0.08 | 14.87±0.05 | 16.77±0.01 | 15.73±0.01 | 13.93±0.08 | 23.20±0.16 |
| | $\beta$=1.0 | 28.93±0.01 | 15.63±0.03 | 19.10±0.14 | 15.30±0.05 | 15.43±0.05 | 22.30±0.46 |
| | #C=1 | 11.00±0.01 | 10.30±0.00 | 10.23±0.00 | 10.30±0.00 | 10.33±0.00 | 10.00±0.00 |
| | #C=2 | 20.40±0.03 | 11.37±0.04 | 11.67±0.06 | 11.00±0.02 | 11.80±0.06 | 10.40±0.00 |
| | #C=3 | 22.30±0.03 | 12.23±0.04 | 14.37±0.13 | 14.10±0.14 | 14.00±0.12 | 18.50±0.14 |
| MNIST | $\beta$=0.01 | 32.20±0.50 | 9.53±0.00 | 9.37±0.00 | 9.00±0.01 | 9.40±0.00 | 9.53±0.00 |
| | $\beta$=0.05 | 60.60±0.07 | 20.80±0.13 | 35.17±0.66 | 35.10±0.87 | 34.13±0.91 | 50.37±1.57 |
| | $\beta$=0.1 | 78.07±0.09 | 45.07±0.37 | 43.23±0.10 | 43.83±0.13 | 44.27±0.21 | 65.53±0.85 |
| | $\beta$=0.3 | 85.60±0.17 | 64.40±0.24 | 64.03±0.11 | 64.17±0.09 | 64.07±0.11 | 75.53±0.22 |
| | $\beta$=0.5 | 91.77±0.00 | 79.43±0.13 | 77.37±0.22 | 78.17±0.25 | 77.90±0.30 | 87.93±0.17 |
| | $\beta$=1.0 | 94.70±0.00 | 85.00±0.10 | 85.10±0.06 | 84.40±0.08 | 84.63±0.08 | 89.30±0.03 |
| | #C=1 | 11.87±0.00 | 10.43±0.02 | 10.13±0.01 | 10.13±0.01 | 10.13±0.01 | 9.93±0.00 |
| | #C=2 | 47.93±0.89 | 13.20±0.09 | 16.47±0.21 | 12.97±0.16 | 12.23±0.07 | 32.57±0.26 |
| | #C=3 | 65.97±0.98 | 26.70±2.24 | 31.67±2.60 | 31.63±3.03 | 31.20±3.24 | 53.80±0.09 |
| SVHN | $\beta$=0.01 | 17.00±0.03 | 13.93±0.16 | 16.57±0.15 | 16.27±0.22 | 13.30±0.20 | 13.97±0.17 |
| | $\beta$=0.05 | 20.23±0.05 | 15.40±0.11 | 15.53±0.12 | 15.53±0.12 | 15.53±0.12 | 17.90±1.01 |
| | $\beta$=0.1 | 32.57±0.53 | 15.17±0.18 | 18.37±0.03 | 18.37±0.03 | 18.33±0.03 | 24.20±0.28 |
| | $\beta$=0.3 | 35.47±0.54 | 18.23±0.29 | 20.77±0.02 | 21.63±0.03 | 21.17±0.01 | 29.23±0.03 |
| | $\beta$=0.5 | 41.17±0.01 | 26.07±0.13 | 27.40±0.00 | 26.27±0.00 | 27.80±0.00 | 36.80±0.13 |
| | $\beta$=1.0 | 44.33±0.01 | 30.77±0.01 | 32.27±0.01 | 30.43±0.03 | 31.97±0.00 | 29.47±2.86 |
| | #C=1 | 19.60±0.00 | 10.10±0.03 | 16.60±0.11 | 16.77±0.13 | 15.53±0.12 | 8.90±0.02 |
| | #C=2 | 31.20±0.01 | 11.80±0.33 | 15.77±0.29 | 15.67±0.31 | 15.60±0.32 | 14.00±0.24 |
| | #C=3 | 34.43±0.40 | 8.93±0.01 | 22.03±0.05 | 18.03±0.12 | 17.50±0.17 | 23.57±0.90 |

Table 6: Comparison with various FL algorithms in one round with 20 local epochs settings.

| Dataset | Partition | FedLPA | Fednova | SCAFFOLD | Fedavg | Fedprox | DENSE |
|---|---|---|---|---|---|---|---|
| FMNIST | $\beta$=0.01 | 24.17±1.13 | 11.90±0.07 | 15.33±0.15 | 13.40±0.09 | 10.57±0.00 | 12.67±0.14 |
| | $\beta$=0.05 | 36.97±0.78 | 18.83±0.55 | 19.93±0.11 | 19.37±0.14 | 19.70±0.17 | 33.13±0.47 |
| | $\beta$=0.1 | 41.83±0.02 | 28.13±1.34 | 23.00±0.41 | 24.63±0.75 | 24.10±1.29 | 36.40±0.02 |
| | $\beta$=0.3 | 60.83±0.38 | 42.50±0.12 | 42.83±0.01 | 40.47±0.22 | 40.63±0.11 | 40.67±0.04 |
| | $\beta$=0.5 | 67.80±0.25 | 53.17±0.25 | 55.23±0.67 | 53.27±0.34 | 54.20±0.52 | 64.60±0.45 |
| | $\beta$=1.0 | 75.47±0.03 | 55.47±0.57 | 54.53±0.52 | 53.57±0.46 | 54.73±0.41 | 70.97±0.02 |
| | #C=1 | 14.07±0.02 | 10.43±0.00 | 11.03±0.02 | 10.43±0.00 | 11.13±0.03 | 10.00±0.00 |
| | #C=2 | 29.67±0.37 | 16.83±0.21 | 22.67±0.27 | 23.27±0.24 | 25.43±0.17 | 24.77±0.14 |
| | #C=3 | 34.37±0.79 | 24.93±0.02 | 26.17±0.01 | 27.70±0.02 | 27.70±0.03 | 29.43±0.22 |
| CIFAR-10 | $\beta$=0.01 | 15.13±0.01 | 10.10±0.00 | 12.50±0.11 | 11.90±0.07 | 11.83±0.07 | 10.50±0.00 |
| | $\beta$=0.05 | 23.37±0.01 | 11.33±0.04 | 10.20±0.00 | 10.00±0.00 | 10.77±0.01 | 14.67±0.02 |
| | $\beta$=0.1 | 25.07±0.00 | 11.60±0.05 | 12.33±0.10 | 12.67±0.14 | 13.23±0.21 | 18.50±0.06 |
| | $\beta$=0.3 | 26.00±0.02 | 16.63±0.13 | 13.10±0.01 | 13.87±0.01 | 14.43±0.02 | 24.97±1.13 |
| | $\beta$=0.5 | 30.60±0.00 | 14.20±0.02 | 13.30±0.04 | 13.13±0.03 | 14.23±0.04 | 27.60±0.06 |
| | $\beta$=1.0 | 26.77±0.10 | 17.60±0.01 | 17.50±0.05 | 18.13±0.04 | 18.20±0.01 | 26.07±0.18 |
| | #C=1 | 10.67±0.01 | 10.20±0.00 | 10.23±0.00 | 10.20±0.00 | 10.27±0.00 | 10.00±0.00 |
| | #C=2 | 22.00±0.00 | 12.03±0.08 | 10.23±0.00 | 11.10±0.02 | 11.40±0.04 | 15.33±0.14 |
| | #C=3 | 23.60±0.05 | 11.97±0.03 | 14.93±0.17 | 13.37±0.09 | 13.63±0.09 | 21.17±0.05 |
| MNIST | $\beta$=0.01 | 32.43±0.86 | 11.00±0.06 | 9.30±0.00 | 9.37±0.00 | 10.33±0.02 | 12.40±0.19 |
| | $\beta$=0.05 | 68.73±0.45 | 26.73±0.24 | 37.77±0.68 | 37.70±0.84 | 36.57±0.75 | 62.03±0.54 |
| | $\beta$=0.1 | 71.77±0.20 | 48.57±0.60 | 45.67±0.23 | 46.63±0.23 | 45.83±0.23 | 66.93±0.25 |
| | $\beta$=0.3 | 90.83±0.01 | 68.17±0.33 | 66.90±0.03 | 66.60±0.15 | 66.03±0.21 | 85.37±0.11 |
| | $\beta$=0.5 | 89.43±0.09 | 80.90±0.14 | 76.63±0.13 | 79.57±0.22 | 79.47±0.27 | 86.07±0.22 |
| | 1.0 | 96.17±0.01 | 86.60±0.13 | 85.90±0.14 | 86.03±0.14 | 86.57±0.15 | 88.40±0.00 |
| | #C=1 | 11.47±0.01 | 10.27±0.01 | 10.13±0.01 | 10.13±0.01 | 10.13±0.01 | 9.87±0.00 |
| | #C=2 | 53.37±0.61 | 17.47±0.48 | 20.70±0.58 | 14.77±0.14 | 13.47±0.02 | 43.33±0.10 |
| | #C=3 | 72.27±0.44 | 28.63±1.61 | 32.93±2.76 | 31.40±2.21 | 30.97±2.50 | 44.30±0.62 |
| SVHN | $\beta$=0.01 | 19.03±0.00 | 14.83±0.13 | 9.33±0.02 | 9.30±0.02 | 9.30±0.02 | 18.23±0.03 |
| | $\beta$=0.05 | 26.27±0.19 | 13.37±0.04 | 15.53±0.12 | 15.53±0.12 | 15.57±0.12 | 24.63±0.41 |
| | $\beta$=0.1 | 28.8±0.70 | 17.47±0.05 | 19.33±0.00 | 19.30±0.01 | 19.70±0.06 | 26.63±0.42 |
| | $\beta$=0.3 | 45.03±0.17 | 27.83±0.04 | 26.53±0.05 | 26.90±0.00 | 27.57±0.01 | 38.27±4.74 |
| | $\beta$=0.5 | 48.00±0.21 | 30.80±0.19 | 32.20±0.33 | 30.07±0.21 | 30.97±0.14 | 43.33±0.08 |
| | $\beta$=1.0 | 62.23±0.05 | 49.07±0.25 | 47.83±0.32 | 48.03±0.25 | 48.53±0.10 | 60.03±0.55 |
| | #C=1 | 16.23±0.23 | 9.83±0.03 | 17.07±0.11 | 16.60±0.12 | 15.50±0.12 | 7.70±0.03 |
| | #C=2 | 27.87±0.49 | 11.83±0.32 | 20.57±0.02 | 19.17±0.01 | 16.00±0.27 | 18.53±0.57 |
| | #C=3 | 42.97±0.02 | 14.10±0.12 | 23.70±0.20 | 25.80±0.39 | 23.37±0.00 | 36.73±0.07 |

Table 7: Comparison with various FL algorithms in one round with 50 local epochs settings.

| Dataset | Partition | FedLPA | Fednova | SCAFFOLD | Fedavg | Fedprox | DENSE |
|---|---|---|---|---|---|---|---|
| FMNIST | $\beta$=0.01 | 19.33±0.43 | 10.13±0.00 | 15.87±0.16 | 18.53±0.35 | 12.97±0.15 | 10.70±0.01 |
| | $\beta$=0.05 | 32.70±0.40 | 19.47±0.53 | 24.10±0.02 | 23.93±0.11 | 22.63±0.20 | 31.33±1.34 |
| | $\beta$=0.1 | 40.00±0.01 | 30.40±1.05 | 27.37±0.23 | 25.83±0.35 | 25.50±0.72 | 39.93±1.10 |
| | $\beta$=0.3 | 62.80±0.41 | 43.67±0.01 | 42.50±0.09 | 41.50±0.10 | 42.23±0.11 | 57.80±0.04 |
| | $\beta$=0.5 | 68.27±0.00 | 55.97±0.23 | 55.27±0.38 | 53.95±0.27 | 55.00±0.26 | 63.50±0.11 |
| | $\beta$=1.0 | 73.47±0.27 | 61.20±0.09 | 60.67±0.19 | 60.77±0.12 | 61.40±0.15 | 66.03±0.06 |
| | #C=1 | 13.30±0.03 | 10.50±0.00 | 11.03±0.02 | 10.50±0.00 | 11.87±0.07 | 10.00±0.00 |
| | #C=2 | 27.60±0.02 | 16.37±0.08 | 23.00±0.12 | 18.37±0.38 | 18.60±0.32 | 29.33±0.44 |
| | #C=3 | 38.13±0.01 | 25.47±0.02 | 25.23±0.29 | 26.70±0.05 | 26.40±0.18 | 37.53±0.17 |
| CIFAR-10 | $\beta$=0.01 | 16.23±0.01 | 10.23±0.00 | 12.27±0.07 | 13.07±0.08 | 12.17±0.09 | 10.33±0.00 |
| | $\beta$=0.05 | 17.93±0.06 | 11.00±0.02 | 10.33±0.00 | 10.13±0.00 | 11.20±0.03 | 7.63±0.01 |
| | $\beta$=0.1 | 19.20±0.02 | 13.17±0.09 | 13.63±0.21 | 12.00±0.03 | 12.43±0.06 | 19.13±0.09 |
| | $\beta$=0.3 | 27.57±0.03 | 12.53±0.05 | 12.33±0.02 | 11.93±0.01 | 12.90±0.01 | 26.03±0.52 |
| | $\beta$=0.5 | 27.57±0.29 | 13.47±0.03 | 12.30±0.00 | 12.47±0.02 | 13.47±0.02 | 26.40±0.23 |
| | $\beta$=1.0 | 30.27±0.02 | 15.47±0.12 | 15.30±0.14 | 15.23±0.13 | 15.53±0.10 | 29.17±2.06 |
| | #C=1 | 10.90±0.00 | 10.30±0.00 | 10.30±0.00 | 10.30±0.00 | 10.33±0.00 | 10.00±0.00 |
| | #C=2 | 21.17±0.06 | 10.13±0.00 | 11.93±0.02 | 10.57±0.01 | 11.27±0.01 | 15.87±0.08 |
| | #C=3 | 23.80±0.01 | 12.00±0.06 | 12.07±0.02 | 12.97±0.04 | 11.90±0.02 | 21.53±0.29 |
| MNIST | $\beta$=0.01 | 34.10±0.88 | 10.57±0.02 | 9.50±0.00 | 9.33±0.02 | 10.13±0.01 | 12.53±0.19 |
| | $\beta$=0.05 | 66.23±0.32 | 32.00±0.78 | 39.70±0.50 | 39.60±0.31 | 39.87±0.15 | 56.63±0.65 |
| | $\beta$=0.1 | 72.90±0.27 | 49.17±0.62 | 47.20±0.22 | 47.07±0.23 | 46.30±0.10 | 69.93±0.27 |
| | $\beta$=0.3 | 87.03±0.02 | 68.30±0.33 | 66.40±0.16 | 67.10±0.09 | 66.17±0.22 | 82.47±0.01 |
| | $\beta$=0.5 | 90.43±0.07 | 80.70±0.13 | 78.13±0.19 | 79.37±0.19 | 79.50±0.22 | 88.30±0.05 |
| | $\beta$=1.0 | 94.47±0.04 | 86.73±0.15 | 85.43±0.11 | 86.07±0.12 | 86.20±0.16 | 89.23±0.01 |
| | #C=1 | 11.37±0.01 | 10.17±0.01 | 10.13±0.01 | 10.13±0.01 | 10.10±0.01 | 9.80±0.00 |
| | #C=2 | 71.07±0.02 | 23.53±1.00 | 22.93±0.80 | 22.63±0.99 | 17.53±0.12 | 42.97±0.17 |
| | #C=3 | 76.17±0.38 | 29.60±1.80 | 33.50±2.91 | 32.77±2.27 | 23.40±3.38 | 57.30±0.12 |
| SVHN | $\beta$=0.01 | 19.60±0.00 | 13.93±0.16 | 13.57±0.19 | 9.50±0.03 | 9.27±0.02 | 19.10±0.52 |
| | $\beta$=0.05 | 22.97±0.01 | 14.87±0.12 | 15.83±0.13 | 15.67±0.12 | 14.67±0.14 | 19.97±0.54 |
| | $\beta$=0.1 | 45.83±1.70 | 22.40±0.21 | 22.97±0.12 | 22.47±0.01 | 24.30±0.04 | 41.47±4.63 |
| | $\beta$=0.3 | 36.30±2.02 | 33.90±0.15 | 33.87±0.20 | 33.50±0.13 | 34.43±0.21 | 29.90±0.56 |
| | $\beta$=0.5 | 51.77±0.01 | 39.70±0.19 | 39.93±0.13 | 38.03±0.14 | 38.33±0.12 | 50.10±1.71 |
| | $\beta$=1.0 | 57.97±0.10 | 56.70±0.15 | 54.03±0.28 | 55.33±0.23 | 55.80±0.15 | 47.80+8.91 |
| | #C=1 | 19.37±0.00 | 9.90±0.03 | 16.57±0.12 | 16.53±0.12 | 15.53±0.12 | 10.00±0.00 |
| | #C=2 | 36.93±0.02 | 12.53±0.25 | 20.30±0.07 | 20.70±0.12 | 15.57±0.22 | 40.77±2.21 |
| | #C=3 | 42.43±0.02 | 21.07±0.31 | 29.63±0.16 | 27.10±0.06 | 24.73±0.00 | 38.50±0.60 |

Here, we present experiments similar to those in Table 1 but with different numbers of epochs (10, 20, 50, 100). The performance of our methods outperforms other approaches, as shown in Table 5, Table 6, Table 7, and Table 8. Without tuning the number of local epochs, our method consistently achieves high performance compared to other baselines.

Table 8: Comparison with various FL algorithms in one round with 100 local epochs settings.

| Dataset | Partition | FedLPA | Fednova | SCAFFOLD | Fedavg | Fedprox | DENSE |
|---|---|---|---|---|---|---|---|
| FMNIST | $\beta$=0.01 | 19.17±0.01 | 11.73±0.06 | 16.10±0.18 | 19.00±0.27 | 12.67±0.12 | 10.07±0.00 |
| | $\beta$=0.05 | 36.77±0.51 | 18.07±0.35 | 22.67±0.03 | 22.73±0.04 | 21.20±0.17 | 31.77±1.14 |
| | $\beta$=0.1 | 35.90±0.12 | 32.83±0.58 | 29.87±0.49 | 30.80±0.28 | 29.33±0.60 | 33.23±1.22 |
| | $\beta$=0.3 | 64.07±0.28 | 47.77±0.07 | 42.20±0.01 | 43.33±0.06 | 46.03±0.06 | 60.30±0.17 |
| | $\beta$=0.5 | 68.73±0.10 | 57.03±0.20 | 55.87±0.38 | 56.10±0.31 | 58.60±0.43 | 64.60±0.01 |
| | $\beta$=1.0 | 76.27±0.00 | 65.00±0.05 | 61.67±0.35 | 65.13±0.14 | 65.03±0.14 | 75.80±0.05 |
| | #C=1 | 13.37±0.04 | 10.87±0.02 | 10.47±0.00 | 10.87±0.02 | 13.23±0.21 | 10.00±0.00 |
| | #C=2 | 31.40±1.16 | 20.93±0.23 | 24.97±0.19 | 23.13±0.25 | 21.50±0.29 | 26.30±1.56 |
| | #C=3 | 49.73±0.24 | 26.97±0.00 | 25.57±0.27 | 26.17±0.22 | 25.50±0.12 | 46.87±0.10 |
| CIFAR-10 | $\beta$=0.01 | 16.93±0.01 | 10.33±0.00 | 10.97±0.02 | 9.57±0.41 | 11.10±0.02 | 11.23±0.01 |
| | $\beta$=0.05 | 19.07±0.01 | 12.33±0.11 | 12.50±0.12 | 10.33±0.00 | 12.60±0.13 | 18.63±0.11 |
| | $\beta$=0.1 | 20.80±0.08 | 12.53±0.05 | 10.33±0.00 | 10.67±0.00 | 11.87±0.03 | 24.30±0.05 |
| | $\beta$=0.3 | 28.33±0.00 | 11.63±0.02 | 11.03±0.01 | 11.07±0.00 | 11.70±0.01 | 28.23±0.36 |
| | $\beta$=0.5 | 29.37±0.01 | 12.07±0.01 | 12.13±0.01 | 11.80±0.01 | 13.17±0.01 | 28.90±0.49 |
| | $\beta$=1.0 | 30.57±0.00 | 14.53±0.09 | 13.93±0.01 | 13.97±0.10 | 15.93±0.11 | 29.37±1.73 |
| | #C=1 | 11.03±0.02 | 10.23±0.00 | 10.23±0.00 | 10.23±0.00 | 10.57±0.01 | 10.00±0.00 |
| | #C=2 | 16.70±0.13 | 10.00±0.00 | 12.90±0.03 | 11.00±0.01 | 11.97±0.03 | 13.67±0.03 |
| | #C=3 | 18.87±0.01 | 11.33±0.03 | 10.70±0.00 | 11.77±0.02 | 11.67±0.02 | 15.97±0.10 |
| MNIST | $\beta$=0.01 | 34.10±0.66 | 13.60±0.32 | 9.33±0.00 | 9.30±0.00 | 9.30±0.00 | 16.63±0.33 |
| | $\beta$=0.05 | 72.47±0.07 | 32.30±0.66 | 41.37±0.37 | 38.57±0.35 | 40.70±0.49 | 55.30±1.88 |
| | $\beta$=0.1 | 78.53±0.20 | 48.20±0.39 | 47.87±0.26 | 47.57±0.19 | 46.93±0.04 | 76.47±0.20 |
| | $\beta$=0.3 | 85.83±0.04 | 68.77±0.28 | 67.43±0.11 | 67.13±0.12 | 65.67±0.36 | 84.23±0.08 |
| | $\beta$=0.5 | 89.03±0.12 | 80.53±0.19 | 79.13±0.23 | 79.00±0.28 | 79.50±0.30 | 88.30±0.31 |
| | $\beta$=1.0 | 94.13±0.03 | 86.53±0.09 | 85.87±0.09 | 85.63±0.08 | 86.17±0.14 | 92.57±0.02 |
| | #C=1 | 11.27±0.01 | 10.30±0.02 | 10.10±0.01 | 10.10±0.01 | 10.13±0.01 | 9.93±0.00 |
| | #C=2 | 71.07±0.35 | 21.00±0.61 | 22.47±0.89 | 18.83±0.55 | 14.50±0.12 | 45.47±0.14 |
| | #C=3 | 76.83±0.32 | 29.63±2.43 | 35.17±2.54 | 32.47±3.15 | 29.2±2.22 | 67.33±0.95 |
| SVHN | $\beta$=0.01 | 19.50±0.00 | 13.90±0.16 | 9.37±0.02 | 12.57±0.22 | 11.60±0.09 | 19.10±0.13 |
| | $\beta$=0.05 | 32.90±0.05 | 13.50±0.03 | 16.03±0.15 | 15.90±0.18 | 16.83±0.10 | 25.80±1.64 |
| | $\beta$=0.1 | 36.63±0.27 | 22.37±0.62 | 24.17±0.20 | 24.83±0.07 | 25.93±0.09 | 26.97±0.23 |
| | $\beta$=0.3 | 56.40±0.01 | 35.43±0.10 | 34.40±0.05 | 35.17±0.10 | 34.40±0.07 | 55.67±1.85 |
| | $\beta$=0.5 | 55.63±0.16 | 39.07±0.03 | 40.33±0.05 | 37.47±0.01 | 37.07±0.12 | 55.53±0.62 |
| | $\beta$=1.0 | 65.57±0.01 | 55.87±0.27 | 55.30±0.26 | 54.80±0.19 | 54.17±0.34 | 62.50±0.12 |
| | #C=1 | 16.27±0.22 | 10.33±0.00 | 13.83±0.17 | 15.67±0.11 | 15.63±0.11 | 12.10±0.30 |
| | #C=2 | 41.87±0.01 | 14.80±0.12 | 22.53±0.07 | 20.77±0.06 | 13.87±0.86 | 41.43±1.77 |
| | #C=3 | 48.70±0.02 | 23.50±0.04 | 30.20±0.08 | 29.20±0.10 | 25.30±0.02 | 48.60±0.49 |

## G.4 EXTREME SETTING, 5 CLIENTS

Table 9: Comparison with various FL algorithms in one round when client number is 5.

| Dataset | Partition | FedLPA | Fednova | SCAFFOLD | Fedavg | Fedprox | DENSE |
|---|---|---|---|---|---|---|---|
| FMNIST | $\beta$=0.01 | 48.13±0.28 | 26.03±0.07 | 30.77±0.49 | 30.80±0.34 | 17.83±0.07 | 44.23±0.14 |
| | $\beta$=0.05 | 55.20±0.17 | 23.40±0.16 | 30.80±0.67 | 29.90±0.12 | 20.43±0.16 | 46.17±0.09 |
| | $\beta$=0.1 | 59.27±0.12 | 33.47±0.16 | 37.77±0.45 | 35.43±0.86 | 32.57±0.98 | 58.73±0.15 |
| | $\beta$=0.3 | 73.13±0.00 | 53.13±0.42 | 52.57±0.46 | 52.03±0.59 | 49.90±0.33 | 63.40±0.06 |
| | $\beta$=0.5 | 74.17±0.02 | 60.27±0.53 | 60.13±0.57 | 59.97±1.14 | 61.67±0.35 | 72.03±0.05 |
| | $\beta$=1.0 | 75.30±0.0 | 63.00±0.05 | 60.87±0.24 | 62.63±0.05 | 60.37±0.01 | 74.93±0.04 |

When the number of clients is set to 5, the experimental results for the FMNIST dataset are shown in Table 9. These results demonstrate that our framework performs well even in extreme situations when the number of clients is relatively small.

## G.5 EXTREME SETTING, $\beta = 0.001$

Here, we demonstrate that even when $\beta = 0.001$ and with different dataset and local epoch number settings, FedLPA has the potential to aggregate models effectively in extreme situations and produce superior results. These results are presented in Table 10.

## G.6 AGGREGATION VISUALIZATION

Figure 6 displays the visualization aggregation of two heterogeneous Gaussians. In comparison to the Fisher Diagonal (Liu et al., 2021) method and Fedavg, our method aggregates the local clients on the Riemannian manifold while treating the empirical Fisher information matrix as a metric of the parameter space, which is much better.

Table 10: Comparison with various FL algorithms in one round with different epoch numbers and $\beta = 0.001$.

| Dataset | epochs number | FedLPA | Fednova | SCAFFOLD | Fedavg | Fedprox | DENSE |
|---|---|---|---|---|---|---|---|
| FMNIST | 10 | 14.57±0.04 | 10.60±0.01 | 10.53±0.01 | 10.60±0.01 | 13.10±0.01 | 10.00±0.01 |
| | 20 | 15.33±0.04 | 10.13±0.00 | 10.23±0.00 | 10.13±0.00 | 12.87±0.16 | 10.00±0.00 |
| | 50 | 13.77±0.02 | 10.57±0.01 | 10.17±0.00 | 10.57±0.01 | 12.30±0.11 | 10.00±0.00 |
| | 100 | 15.83±0.03 | 10.17±0.00 | 10.73±0.00 | 10.17±0.00 | 13.23±0.21 | 10.00±0.00 |
| | 200 | 14.53±0.00 | 10.07±0.00 | 10.10±0.00 | 10.07±0.00 | 12.50±0.12 | 10.00±0.00 |
| CIFAR-10 | 10 | 11.50±0.00 | 10.27±0.00 | 10.17±0.00 | 10.27±0.00 | 10.33±0.00 | 10.00±0.00 |
| | 20 | 10.57±0.01 | 10.27±0.00 | 10.13±0.00 | 10.27±0.00 | 10.30±0.00 | 10.00±0.00 |
| | 50 | 10.77±0.01 | 10.23±0.00 | 10.33±0.00 | 10.23±0.00 | 10.33±0.00 | 10.00±0.00 |
| | 100 | 10.90±0.01 | 10.20±0.00 | 10.30±0.00 | 10.23±0.00 | 10.57±0.01 | 10.00±0.00 |
| | 200 | 10.87±0.02 | 10.27±0.00 | 10.23±0.00 | 10.27±0.00 | 10.37±0.01 | 10.00±0.00 |
| MNIST | 10 | 24.10±0.17 | 10.07±0.01 | 12.17±0.07 | 11.83±0.05 | 12.17±0.12 | 9.90±0.00 |
| | 20 | 19.53±0.33 | 10.07±0.01 | 12.07±0.07 | 13.37±0.08 | 12.37±0.12 | 9.27±0.00 |
| | 50 | 16.93±0.37 | 10.07±0.01 | 10.80±0.04 | 13.17±0.09 | 13.13±0.25 | 11.40±0.08 |
| | 100 | 19.07±0.41 | 10.13±0.01 | 10.97±0.00 | 11.37±0.02 | 12.90±0.13 | 12.83±0.17 |
| | 200 | 15.63±0.03 | 10.07±0.01 | 11.13±0.06 | 12.50±0.04 | 11.83±0.11 | 9.27±0.00 |
| SVHN | 10 | 17.50±0.02 | 15.90±0.00 | 15.53±0.12 | 15.53±0.12 | 15.53±0.12 | 17.13±0.03 |
| | 20 | 20.10±0.21 | 15.90±0.00 | 15.53±0.12 | 15.53±0.12 | 14.00±0.11 | 17.13±0.03 |
| | 50 | 20.07±0.71 | 16.30±0.00 | 15.50±0.21 | 15.13±0.16 | 14.03±0.07 | 15.17±0.16 |
| | 100 | 19.70±0.00 | 15.90±0.00 | 15.10±0.16 | 15.53±0.12 | 13.77±0.10 | 18.47±0.05 |
| | 200 | 19.13±0.00 | 13.90±0.11 | 14.90±0.19 | 15.13±0.16 | 13.27±0.06 | 15.23±0.16 |

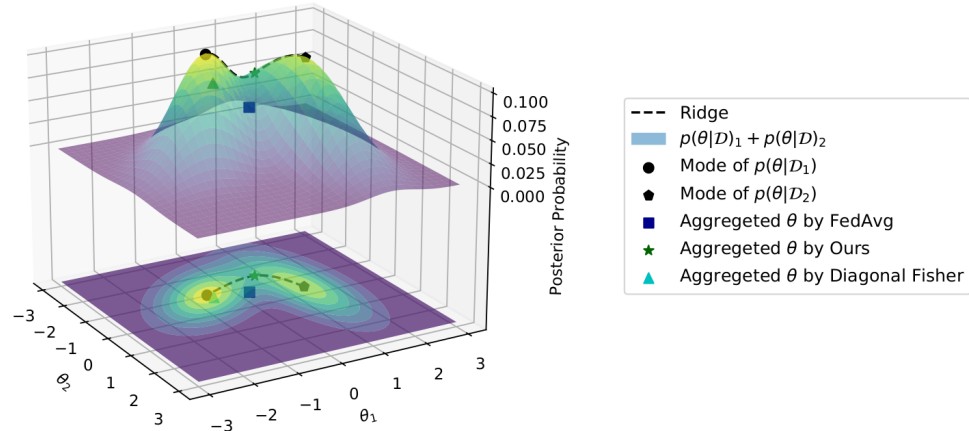

Figure 6: Aggregation Visualization.

## G.7 EXPERIMENTS WITH FEDOV

We compare with FedOV[2], the state-of-the-art method which addresses label skews in one-shot federated learning. We run the experiments with fair comparison (same model size) on MNIST dataset with #C=2 partition setting. Table 11 shows that our method could be comparable with FedOV in some scenarios even when FedOV transmits the unknown label information to the clients and utilizes the knowledge distillation. As the epoch number of local clients equals to 50,100,200, FedLPA outperforms FedOV.

---

[2]https://github.com/Xtra-Computing/FedOV

Table 11: Comparison with FedOV on MNIST with #C=2.

| epoch number | 10 | 20 | 50 | 100 | 200 |
|---|---|---|---|---|---|
| FedLPA | 47.93±0.89 | 53.37±0.61 | 71.07±0.02 | 71.07±0.35 | 69.63±0.29 |
| FedOV | 71.0±0.25 | 70.27±0.39 | 69.23±0.31 | 65.83±0.23 | 64.50±0.38 |

Table 12: Running time and computation overhead evaluation.

| FedLPA | Fednova | SCAFFOLD | Fedavg | Fedprox | DENSE | FedOV |
|--------|---------|----------|--------|---------|-------|-------|
| 65mins | 50mins | 50mins | 50mins | 75mins | 400mins | 150mins |

### G.8 COMMUNICATION OVERHEAD EVALUATION

Table 4 shows the communication overhead evaluation of a simple CNN with 5 layers on CIFAR-10 dataset. The results are given based on the experiments. In this section, we will give a concrete example to show the details.

The communication bits are the number of bits that are transmitted between a server and a client in a directed communication. It reflects the communication efficiency of federated learning algorithms. Better algorithms should have lower communication bits.

The default floating point precision is 32 bits in Pytorch. We use a fully-connected neural network model with architecture 784-254-64-10 as an example to show the calculation, which has $784*256 + 256 + 254*64 + 64 + 64*10 + 10 = 217930$ floating point numbers, which is 6973760 bits or around $0.831$ MB.

For a single directed communication from a client to the server or vice versa, the cost for Fedavg, Fedprox, Fednova, FedPA, and DENSE is $0.831$ MB each. SCAFFOLD costs $1.662$ MB for the same communication, which is double the amount of the others.

For a single communication from a client to the server, our method requires additional upload of $\mathbf{A}_k$ and $\mathbf{B}_k$, which contain $785*785 + 256*256 + 257*257 + 64*64 + 65*65 + 10*10 = 756231$ floating point numbers in total. Note, as $\mathbf{A}_k$ and $\mathbf{B}_k$ are symmetric matrices, we only need to upload the upper triangular part of them, reducing the total to roughly $756231/2 = 378115.5$ floating point numbers as about $1.442$ MB. Therefore, our approach costs $2.272$ MB for the one directed communication, which is $2.734$ times as Fedavg, Fedprox, and DENSE, and $1.367$ times as SCAFFOLD.

In practical real-world applications, convolutional neural networks (CNNs) are commonly used instead of fully connected networks, and our approach introduces relatively low extra communication overhead, as indicated in Table 4. Specifically in Table 4, our approach incurs about 1.13 times the communication overhead of Fedavg, Fednova, Fedprox, and DENSE, while being only 0.56 times the overhead of SCAFFOLD.

However, as Figure 2 demonstrates, to achieve the same performance as FedLPA, Fedavg, Fednova, SCAFFOLD, and Fedprox require more communication rounds, resulting in a heavier data transfer burden on the system.

### G.9 RUNNING TIME AND COMPUTATION OVERHEAD EVALUATION

The running times of different algorithms, using a simple CNN on the CIFAR-10 dataset, are summarized in Table 12. In this experiment, there are 10 clients, each running 200 local epochs with only one communication round. Our device is a single 2080Ti GPU. Compared to the state-of-the-art methods FedOV and DENSE, our method is efficient and slightly slower than the fastest algorithm. Notably, DENSE consumes almost 7 times the computational resources, as the knowledge distillation method is computationally intensive and resource-demanding. It's important to note that while our method is efficient, it also yields the best results.

### G.10 ARTIFACT DETAILS

We have uploaded the codebase containing all the methods compared in our paper. Setting up the environment is relatively straightforward with the provided readme file. If you refer to the scripts folder, you will find all the bash scripts necessary to reproduce the tables and figures from our experiments.

The experiments.sh script covers the experiments in Table 1, Table 5, Table 6, Table 7, and Table 8. Running these experiments on a single 2080Ti GPU will take approximately 81 days. Specifically, Table 1 itself will take about 35 days.

The experiments_client.sh script covers the experiments in Table 2, requiring approximately 40 days on a single 2080Ti GPU.

The experiments_coor.sh script covers the experiments in Table 3, which can be completed in 2 days.

The experiments_extreme.sh script reproduces the experiments in Table 10 and takes about 10 days.

The experiments_extreme_clients.sh script covers the experiments in Table 9 and requires approximately 4 days of GPU processing.

Running experiments_multiple_round.sh will yield the results as shown in Figure 2, and this process takes about 1 day.

To generate the t-SNE visualizations shown in Figure 1, Figure 4, and Figure 5, you can use the experiments.py script with the "alg=tsne" option.

