# OpenReview forum: "FedLPA: Personalized One-shot Federated Learning with Layer-Wise Posterior Aggregation"
_ICLR.cc/2024/Conference — Submitted to ICLR 2024_

### Official Review · Reviewer_8Erm · 2023-10-31

**Soundness:** 3 good
**Presentation:** 2 fair
**Contribution:** 2 fair
**Rating:** 5
**Confidence:** 3

**Summary:**

This work proposes a one-shot FL method through using approximated local posteriors of heterogeneous clients. The key idea is to use empirical Fisher to approximate the local inverse covariance (which is utilized in the final aggregation). By using local statistics authors are able to do the aggregation step while taking the heterogeneity of the clients into account.

**Strengths:**

- The proposed method is sound, the way the authors want to utilize the local covariance's and use empirical Fisher for approximation is a good idea to employ in heterogeneous settings.
- In the reported experiments the method significantly outperforms competing methods.
- The first 2 sections are well-written and motivates the work decently.

**Weaknesses:**

- One of the major weaknesses is the way the methodology is presented. In Section 3 various subsections are sequentially presented; but the subsections are not connected well, it is hard to follow the sequence of methodology through text. The algorithm should be put into main text, and well connected to the sections. I would suggest rewriting the algorithm in more details and possibly adding a figure as an overview.
- The authors call their method 'personalized'. In personalized FL the inference is made through individual models of the clients. But, as far as I understand, the output of your methodology is a single global model that would hopefully work for every client; although you obtain local posteriors they are just intermediators for the global model training. If this is the case, this is not a personalized method but a heterogeneous FL method. Otherwise, please clarify.
- I think the current experiments are not enough to show the efficacy of the method. In particular, majority of the experiments are done with 20 clients, which is small for the FL. Also, there is no comparison to local only training.
- More experiments with higher number of clients should be added to text if possible. Also the performance change w.r.t number of clients and data samples should be reported.
- Lack of experiments on synthetically generated data is not desirable for such a statistically motivated method.

**Questions:**

- In section 3.5 is there a specific reason to introduced the l2 squared loss other than available optimization tools?
- See above for other points.

---

> ### Author Response · Authors · 2023-11-20
> **Response to Reviewer 8Erm-1**
>
> We kindly thank the reviewer for their constructive and valuable feedback. We appreciate the time and effort you have dedicated to assessing our work and we are happy to address the concerns below. We sincerely hope you can read our responses and adjust the scores if appropriate.
>
> Q:One of the major weaknesses is the way the methodology is presented. In Section 3 various subsections are sequentially presented; but the subsections are not connected well, it is hard to follow the sequence of methodology through text. The algorithm should be put into main text, and well connected to the sections. I would suggest rewriting the algorithm in more details and possibly adding a figure as an overview.
>
> A: Thanks for your suggestion. We will improve our logic of Section 3 and in the camera-ready version, we will rewrite the algorithm in more details and add a figure as an overview. Our logic of the subsections in Section 3 are as follows:
>
> In Section 3.1, we want to show that a global objective function could be obtained via the local posterior.
>
> In Section 3.2, we discuss how to get the local posterior and we convert this problem into approximate Hessian. The most efficient method to get the Hession is using the empirical Fisher information matrix.
>
> In Section 3.3, we show how to compute the Block-Diagonal empirical Fisher information matrices, which is better than the empirical Fisher information matrix to approximate Hessian.
>
> In Section 3.4, with the Block-Diagonal empirical Fisher information matrices from local models, we show how to aggregate these matrices.
>
> In Section 3.5, as the aggregation is unacceptable, we propose the efficient and practical method to do the train the parameters of the global model.
>
> We will add the logical sentences between subsections and more explanations in the corresponding subsections in the camera-ready version.
>
> Q: The authors call their method 'personalized'. In personalized FL the inference is made through individual models of the clients. But, as far as I understand, the output of your methodology is a single global model that would hopefully work for every client; although you obtain local posteriors they are just intermediators for the global model training. If this is the case, this is not a personalized method but a heterogeneous FL method. Otherwise, please clarify.
>
> A: Thanks for the comments on the aspect of the personalization. We understand that the personalized federated learning setting may contain the multiple aspects. In this paper, we mainly consider that each client has a different biased dataset, which is one of the personalized federated learning settings. We agree that our claim of the personalization is somehow weak. We will weaken the claim of the personalization in the camera-ready version.
>
> Q: I think the current experiments are not enough to show the efficacy of the method. In particular, majority of the experiments are done with 20 clients, which is small for the FL. Also, there is no comparison to local only training.
>
> A: In Section 4.3, we show that the experimental results of varying number of clients on FMNIST dataset. We conduct the experiments with 50 clients, which is also the maximal number of clients shown in other federated one-shot learning papers[1][2]. In our G.10, we show that the computing resources we need to execute the experiments of this paper. Due to the resource limit, we also choose 50 as our maximal number.
>
> For the comparison to local only training, we have shown them in our paper in Fig. 1 in Section 3.6 and the corresponding details in Appendix G.2.
>
> [1]Su, S., Li, B., & Xue, X. (2023). One-shot Federated Learning without server-side training. Neural Networks, 164, 203-215.
>
> [2]Heinbaugh, Clare Elizabeth, Emilio Luz-Ricca, and Huajie Shao. "Data-Free One-Shot Federated Learning Under Very High Statistical Heterogeneity." The Eleventh International Conference on Learning Representations. 2022.

---

> ### Author Response · Authors · 2023-11-20
> **Response to Reviewer 8Erm-2**
>
> Q:More experiments with higher number of clients should be added to text if possible. Also the performance change w.r.t number of clients and data samples should be reported. Lack of experiments on synthetically generated data is not desirable for such a statistically motivated method.
>
> A: Thanks for the suggestion. We have added the experiments with our method on the same experiment setting in the paper with three random seeds with 10 clients. We conducted experiments on FMNIST datasets with beta=0.1, 0.3 and 0.5. The performance change w.r.t number of data samples are shown as follows:
>
> | **Accuracy (Beta=0.1)** | **Data sample proportions**   |
> |---------------|------------------------|
> | 55.33±0.06       |100%|
> | 53.88±1.14       | 80%|
> | 53.15±0.82       | 60%|
> | 53.20±0.21       | 40%|
> | 45.71±0.13       | 20%|
>
> | **Accuracy (Beta=0.3)**     | **Data sample proportions**   |
> |---------------|------------------------|
> | 68.20±0.04       |100%|
> | 65.47±0.02       | 80%|
> | 64.80±0.71       | 60%|
> | 64.10±0.40       | 40%|
> | 62.15±0.03       | 20%|
>
> | **Accuracy (Beta=0.5)**     | **Data sample proportions**   |
> |---------------|------------------------|
> | 73.33±0.06       |100%|
> | 73.17±0.05       | 80%|
> | 72.40±0.29       | 60%|
> | 70.02±0.17       | 40%|
> | 68.54±2.02       | 20%|
>
> With the above tables, we could see that our method FedLPA could yield satisfactory results even with only 20% data samples under multiple settings. Due to the rebuttal time limit, we have not added the synthetically generated data experiments. We believe that our proposed FedLPA works well on the synthetically generated dataset since it already performs well on the realistic datasets.
>
> Q: In section 3.5 is there a specific reason to introduce the l2 squared loss other than available optimization tools?
>
> A: We choose the  l2 squared loss because it is the most common convex function used in the machine learning field and we have also proved the converge rate of FedLPA in l2 squared loss.  We agree that other convex loss functions will not affect the implementation of FedLPA.  We leave the impact of other loss function as future work.

---

> ### Author Response · Authors · 2023-11-22
> **Looking forward to your feedbacks for our rebuttal**
>
> Dear Reviewer 8Erm,
>
> We'd appreciate it if you could provide any feedback and add further comments based on our rebuttals . We are willing to provide further quick reponses. Thanks for your time and efforts!

---

### Official Review · Reviewer_yUMy · 2023-10-31

**Soundness:** 3 good
**Presentation:** 3 good
**Contribution:** 3 good
**Rating:** 6
**Confidence:** 3

**Summary:**

This paper introduces a one-shot federated learning approach under a Bayesian framework. This approach solves the communication burden in transmitting the Hessian of log posterior by approximating it with the empirical Fisher information matrix and further approximating the empirical Fisher into layer-wise block-diagonal matrix by assuming layer independence. The layer-wise block-diagonal matrix is then decomposed into smaller factor matrices with Kronecker-factored approximation. These approximation methods combined greatly saved the communication cost than naively transmitting the full matrix. The author considered a thorough list of baselines to compare and showed that the proposed FedLPA outperformed those methods in a one-shot setting.

**Strengths:**

- This paper is well-written with methods clearly explained despite its complexity.
- The proposed method is novel and technically solid. The different approximation steps are driven with empirical constraints in Federated learning. Though most of the linear algebra tricks are based on existing works, they are applied in an innovative way to solve the focused one-short FL problem.
- The empirical evaluation is thorough with multiple baseline methods implemented and compared in different heterogeneous settings. The results in the one-shot setting demonstrates that the proposed methods are promising.

**Weaknesses:**

- The proposed method is composed of multiple approximations: 1) empirical fisher to approximate the Hessian 2) block-diagonal Fisher matrix instead of full, 3) approximating global model parameter $\bar{M}$ with optimization problem in Equation 14. However, there is no ablation to understand how each approximation step impacts the final results.
- FedLPA requires transmitting individual (instead of aggregation) $A_k, B_k, M_k$ in order to solve the optimization problem in Equation 14. Exposing individual statistics to the server can have privacy concerns and cannot be compatible with standard secure aggregation protocol or central differential privacy methods.

**Questions:**

- Why are multi-round results of FedLPA worse than the alternatives as shown in Figure 2?
- Consider a large model where one single layer weight can be enormous, how would one further decompose its Fisher information for communication efficiency?

---

> ### Author Response · Authors · 2023-11-20
> **Response to Reviewer yUMy-1**
>
> We kindly thank the reviewer for their constructive and valuable feedback. We appreciate the time and effort you have dedicated to assessing our work and we are happy to address the concerns below. We sincerely hope you can read our responses and adjust the scores if appropriate.
>
> Q:The proposed method is composed of multiple approximations: 1) empirical fisher to approximate the Hessian 2) block-diagonal Fisher matrix instead of full, 3) approximating global model parameter $\bar{M}$ with optimization problem in Equation 14. However, there is no ablation to understand how each approximation step impacts the final results.
>
> A:
>
> (1) Empirical Fisher to approximate the Hessian:
>
> Although empirical Fisher has been successfully applied in many methods and yielded good results, discussions concerning the approximation error of empirical Fisher are limited. Fortunately, previous work [1] provides a detailed critical discussion of the empirical Fisher approximation.
>
> i. Fisher to approximate the Hessian:
>
> When the loss function represents an exponential family distribution, the Fisher is a well-justified approximation of the Hessian, and its approximation error can be bounded in terms of residuals. The accuracy of this approximation improves as the residuals diminish and is exact when the data is perfectly fitted.
>
> ii. Empirical Fisher to Fisher:
>
> It's noted that the Fisher and empirical Fisher coincide near minima of the loss function under two conditions:
>
>   A. The model distribution closely approximates the data distribution.
>
>   B. A sufficiently large number of samples allow both the Fisher and empirical Fisher to converge to their respective average values in the population.
>
> In practical environments, especially condition 1, might not hold, causing bias between empirical Fisher and Fisher. However, empirical Fisher still contains effective covariance information. In second-order optimization methods, the covariance information in empirical Fisher can adapt to the gradient noise in stochastic optimization. Nevertheless, referencing [3], we can use the model’s predictive distribution to obtain an unbiased estimate of the true Fisher at the same computational cost as empirical Fisher.
>
> (2) Block-diagonal Fisher matrix to approximate the full one:
>
> Paper [2] provides a detailed evaluation and testing of using block-diagonal Fisher to approximate the full one.
> Firstly, Chapter 6.3.1 "Interpretations of this approximation" in the paper indicates that using a block-wise Kronecker-factored Fisher closely approximates the full Fisher. Although there is a bias term (due to the approximation in our Appendix Equation 27), this term approximates zero when there are sufficient samples. Furthermore, the paper examines the approximation quality of block-diagonal Fisher compared with the true Fisher and suggests that block-diagonal Fisher captures the main correlations, while the remaining correlations have a minimal impact on the experimental results.
>
> (3)Besides, we have added some experiments for more ablation studies with our method on the same experiment setting in the paper with three random seeds with 10 clients. We conducted experiments on FMNIST datasets with beta=0.1, 0.3 and 0.5. The results are shown as follows:
>
>
> |**Number of iteration**| **Accuracy (Beta=0.1)** | **Accuracy (Beta=0.3)**  | **Accuracy (Beta=0.5)**  |
> |---------------|---------------|------------------------|------------------------|
> |1000  | 52.81±0.71       |60.31±0.23|72.11±0.57|
> |5000  | 59.70±0.32       |68.09±0.30|74.27±0.12|
> |10000| 55.33±0.06       |68.20±0.04|73.33±0.06|
> |20000| 58.41±0.05       |68.11±0.07|73.51±0.02|
>
> With the experiment results, we could know that 5000 iterations is enough to get the ideal results.
>
> In Appendix F, we also show the linear convergence rate for Eq. 14.
>
> Additionally, it's worth noting that concerning Laplace approximation, the analysis in [4] suggests that the error of Laplace approximation is inversely proportional to the input dimension 'n' with $O(n^{-1})$. According to this conclusion, it can be inferred that in our method, for each layer of the neural network, the error of Laplace approximation is inversely proportional to its width. When the neural network is infinitely wide, the approximation error tends towards zero.
>
> We will add the above explanations into Appendix in the camera-ready version.
>
> [1]Kunstner F, Hennig P, Balles L. Limitations of the empirical fisher approximation for natural gradient descent. Advances in neural information processing systems.
>
> [2]Martens, James, Jimmy Ba, and Matt Johnson. "Kronecker-factored curvature approximations for recurrent neural networks." International Conference on Learning Representations. 2018.
>
> [3]Martens J. Second-order optimization for neural networks. University of Toronto, 2016.
>
> [4]Bilodeau, Blair, Yanbo Tang, and Alex Stringer. "On the tightness of the Laplace approximation for statistical inference." Statistics & Probability Letters 2023.

---

> ### Author Response · Authors · 2023-11-20
> **Response to Reviewer yUMy-2**
>
> Q:FedLPA requires transmitting individual (instead of aggregation) $A_k$, $B_k$, $M_k$ in order to solve the optimization problem in Equation 14. Exposing individual statistics to the server can have privacy concerns and cannot be compatible with standard secure aggregation protocol or central differential privacy methods.
>
> A: Thanks for proposing the privacy concern. In the FedLPA, $A_k$ is computed via the activations while $B_k$ is computed via the linear pre-activations of the layer. We note that $A_k$, $B_k$ and $M_k$ do not carry any label information, thus the transmission of $A_k$, $B_k$ and $M_k$ will not leak any label privacy. Otherwise, we agree that $A_k$, $B_k$ and $M_k$ are a function of data which may contain privacy-sensitive information of the local training data. However, our proposed FedLPA has the same privacy-preserving level as the conventional federated learning algorithms (i.e, FedAvg, FedProx, FedNova and Dense), which are all vulnerable to some privacy attacks (e.g, membership inference attack or reconstruction attacks). We believe that combining differential privacy with FedLPA can enhance the privacy protection of the training data, which is the most promising future direction. However, in this paper, we mainly focus on the efficiency and accuracy of one-shot federated learning algorithms and propose FedLPA which outperforms the SOTA one-shot algorithms on the efficiency and accuracy.
>
>
> Q:Why are multi-round results of FedLPA worse than the alternatives as shown in Figure 2?
>
> A: As we shown in our Appendix D.1, we give an example, showing that the block-diagonal empirical Fisher information matrix could well capture the coarse structure of the full Fisher information matrix based on a neural network with three convolution layers and one fully connected layer trained on MNIST dataset. The four diagonal blocks corresponding to four layers, can capture the most useful information of the original Fisher Information Matrix. Thus, the approximation of the block-diagonal empirical Fisher information matrix can converge fast with most information learned and works well in one-shot scenario. However, it neglects inferior information in non- diagonal blocks.
>
> When we extend FedLPA to the multi-round setting, the inferior information that we omit may enrich the learning performance. While in the one-shot setting, the most useful information of the original Fisher Information Matrix matters.
>
>
> Q: Consider a large model where one single layer weight can be enormous, how would one further decompose its Fisher information for communication efficiency?
>
> A: If a layer's weight in a large model has a considerably high dimension, implying a Fisher matrix with a large dimension, it significantly increases communication costs. In such cases, the most intuitive approach is to explore the possibility of dimensionality reduction for its Fisher matrix.
>
> A promising approach to enhance the efficiency of our method may employ some low-rank factorization techniques [5]. As described in [6], the main idea involves performing an eigendecomposition on the Kronecker factors in [7], while preserving only the eigenvectors corresponding to the top k largest eigenvalues. As a result, this approach drastically reduces space complexity, enabling communication costs to be compared favorably with diagonal Fisher matrices.
>
> [5]Lee, Jongseok, et al. "Estimating model uncertainty of neural networks in sparse information form." International Conference on Machine Learning. PMLR, 2020.
>
> [6]Daxberger, Erik, et al. "Laplace redux-effortless bayesian deep learning." Advances in Neural Information Processing Systems 34 (2021).
>
> [7]George, Thomas, et al. "Fast approximate natural gradient descent in a kronecker factored eigenbasis." Advances in Neural Information Processing Systems 31 (2018).

---

> ### Author Response · Authors · 2023-11-22
> **Looking forward to your feedbacks for our rebuttal**
>
> Dear Reviewer yUMy,
>
> We'd appreciate it if you could provide any feedback and add further comments based on our rebuttals . We are willing to provide further quick reponses. Thanks for your time and efforts!

---

### Official Review · Reviewer_dmu6 · 2023-10-31

**Soundness:** 2 fair
**Presentation:** 1 poor
**Contribution:** 1 poor
**Rating:** 3
**Confidence:** 5

**Summary:**

The paper proposes FedLPA, a method for one-round aggregation in federated learning. Motivated by a posterior inference view of FL, FedLPA proposes to use the (Kronecker-factored) Fisher information of the local models at clients for aggregation. Given clients send their local Fisher information, the server then optimizes a quadratic objective to get the weights of the global model. Experiments are conducted on MNIST, FMNIST, SVHN and CIFAR-10 to demonstrate the improvement offered by FedLPA across various settings of heterogeneity along with other ablation studies.

**Strengths:**

Experimental results show that the proposed FedLPA can outperform vanilla averaging (FedAvg) and distillation-style based aggregation (DENSE) across a range of standard datasets, especially when the data heterogeneity is high. Also while FedLPA does increase overall computation as shown in Table 4, it is still much less than DENSE.

**Weaknesses:**

**1. Writing.** There are numerous writing issues throughout the paper. Many sentences are grammatically incorrect, have incorrect punctuation and/or simply don't make sense. Some words are incorrectly capitalized or not capitalized when they should be.  I am highlighting a few such cases below.

* Page 1: "With the primary objectives of safeguarding data privacy and curbing the aggregation and management of data across institutions, the distribution of data exhibits variations among clients" -> The text before and after the comma are completely disconnected.

* "Fedavg" should be FedAvg everywhere in the paper

* Page 2: "Layer-wise" -> layer-wise

* Page 2: "Fisher information matrices as a metric of the parameter space" -> Metric to measure what? How is the parameter space defined?

* Page 2: " multi-variates linear objective function and using its quadratic form" -> I don't understand what the authors mean by quadratic form of a linear objective

* Page 2: "Nevertheless, from the theoretical analysis, we show that FedLPA has a linear convergence rate" -> Why is there a nevertheless here? Given that the objective is quadratic, gradient descent should have a linear convergence rate.

* Page 4: "globally variational inference using Eq. 2" -> "global variational inference using Eq. 2"

* Page 4: "uploads probability parameters to the server" -> What are probability parameters?

* Page 5: " Modern algorithms (Rumelhart et al., 1986; Martens & Grosse, 2015a) allow the local training process to obtain an optimal, regarded as the expectation $\mu_k$ in the above equations"-> This is completely grammatically incorrect. Please fix this sentence.

* Page 5: "it is an approximate of the Fisher information matrix" -> " it is an approximation of the Fisher information matrix"

* Page 5: computing all co-relations is impossible, which are inaccurate" ->  Text before and after the comma is disconnected.



**2. Mathematical definitions are not precise.** To add to the writing issues, in many places the mathematical notation/assumptions are not properly defined. Some examples are given below
* The second proportionality in Eq. (2) holds only under the flat prior assumption, i.e. $p(\theta) \sim 1$. The authors have not stated this clearly.
* In the line with the definition of Kronecker product (just below Eq. (10)), the authors are missing an expectation in the definition of $A_{kl}$ and $B_{kl}$.
* Section 3.5: What is $\bar{\Sigma}$ and $\bar{z}$ ? In Section 3.4, the authors have just defined $\bar{\Sigma}_l$ and $\bar{z}_l$ for a layer $l$.
* Section 3.5: "optimal solution of $\bar{\mu}$" -> "optimal solution of $f(\bar{\mu})$". Optimal solution of a vector does not make sense.
* Section 3.5: "As we have $\mu = \bar{\Sigma}\cdot \bar{z}$"- > the authors already defined $\bar{\mu} = vec(\bar{M})$ earlier in the same line. I'm not sure what re-definition is doing.


**3. No connection with personalization.** The title of the paper states "Personalized One Shot Federated Learning". The authors also write about the benefits of personalization in paragraph 4 of the Introduction section. However, the rest of the paper just focuses on the server learning a single global model. I don't understand what is the connection with personalization here.

**4. Limited novelty, missing references, privacy concerns.** The idea of formulating FL as a posterior inference problem is already well-known (Al Shedivat et al. 2020) and the idea of approximating the Hessian as the Fisher is also standard  (Ritter et al. 2018). Moreover, the resulting concept of using the Fisher information to aggregate models has already been well-explored in previous work [1], which the authors have not cited. The only difference here is that FedLPA proposes to use the K-FAC while [1] uses the diagonal Fisher. However, I feel this is not significant enough novelty. Moreover, implementing the K-FAC requires clients to send $A_k, B_k, M_k$ matrices which increases communication compared to just the diagonal Fisher. The authors also claim without any justification that $A_k,B_k, M_k$ preserve the data/label privacy.  This claim has to be supported by empirical evidence/theory in order for it to be justified.



**5. Weak baselines and easy datasets.** Among the baselines, I understand the comparison with FedAvg to show improvement over simple averaging. However SCAFFOLD, FedNova and FedProx are not good one-shot baselines. These methods are primarily designed for multi-round FL and therefore I am not surprised to see that their performance is similar to FedAvg in the experiments. DENSE is the only one-shot basline which can be considered competitive. The authors have cited other one-shot algorithms like FedOV and FedCAVE but have not compared to them based on the argument that they entail sharing more client side information. However, I think it fair to compare with these algorithms since FedLPA also requires clients to share and $A_k$ and $B_k$ matrices.

FedLPA seems to outperform baselines significantly only for easier datasets such as FMNIST, MNIST and SVHN. For CIFAR-10 (which can be considered as the hardest dataset), the improvement over DENSE seems to reduce. Therefore I was interested in seeing the performance of FedLPA for even harder datasets such CIFAR-100 and Tiny-ImageNet as done in the DENSE paper. In addition, the paper would be significantly strengthened if the authors considered more realistic FL datasets such as EMNIST and Shakespeare [2].

**Questions:**

Please see my comments in Weakness 1 and 2 to improve the writing of the paper and suggestions for more experiments in Weakness 5. In addition, I was curious to know why the computation cost of Fedprox is higher than FedAvg and FedNova in Table 4.


**References**

[1] Matena, Michael S., and Colin A. Raffel. "Merging models with fisher-weighted averaging." Advances in Neural Information Processing Systems 35 (2022): 17703-17716.

[2] Reddi, Sashank, et al. "Adaptive federated optimization." arXiv preprint arXiv:2003.00295 (2020).

---

> ### Author Response · Authors · 2023-11-20
> **Response to Reviewer dmu6 (1/5)**
>
> We kindly thank the reviewer for their constructive and valuable feedback. We appreciate the time and effort you have dedicated to assessing our work and we are happy to address some of the concerns below. We sincerely hope you can read our responses and adjust the scores if appropriate.
>
>
> Q: Page 1: "With the primary objectives of safeguarding data privacy and curbing the aggregation and management of data across institutions, the distribution of data exhibits variations among clients" -> The text before and after the comma are completely disconnected.
>
> A: Thanks for the doubt on the inner logic of the above sentence. Here, we want to show that in order to safeguard the data privacy, the conventional federated learning algorithm will use the aggregation methods and follows the data management rules of different institutions. However, in this way, the model are not trained with full data, which will incurs the problem that ‘the distribution of data exhibits variations among clients’. We will add the above explanations in the camera-ready version.
>
>
> Q:"Fedavg" should be FedAvg everywhere in the paper.
>
> A: Thanks for notifying our typos. We will revise all the “Fednova”, “Fedavg”, “Fedprox” in the camera-ready final version.
>
> Q: Page 2: "Fisher information matrices as a metric of the parameter space" -> Metric to measure what? How is the parameter space defined?
>
> A: FedLPA will consider the parameter space to aggregate the local models. Bear on that thought, we use FedLPA to aggregate the posteriors of local models using the accurately computed block-diagonal empirical Fisher information matrices.
>
> Q: Page 2: " multi-variates linear objective function and using its quadratic form" -> I don't understand what the authors mean by quadratic form of a linear objective
>
> A: In the Page 6 Section 3.5, Eq. 14 is our final multi-variates linear objective function and it is in the quadratic form.
>
> Q: Page 2: "Nevertheless, from the theoretical analysis, we show that FedLPA has a linear convergence rate" -> Why is there a nevertheless here? Given that the objective is quadratic, gradient descent should have a linear convergence rate.
>
> A: In Appendix F, we have shown the linear convergence rate. We will revise the sentence into “Furthermore, from the theoretical analysis” in the camera-ready version.
>
> Q:"uploads probability parameters to the server" -> What are probability parameters?
>
> A: The data distribution may be biased on different dataset and each client may have a different number of different data samples. The probability parameters are the proportion of the number of data samples on each client over the total number of all the data samples. We will add the above explanations in the camera-ready version.
>
> Q: Page 5: computing all co-relations is impossible, which are inaccurate" -> Text before and after the comma is disconnected.
>
> A: Here, we want to denote that“which are inaccurate” means that the methods of Kirkpatrick et al. (2017); Liu et al. (2021) in the paper are not accurate, as they could not compute all co-relations. We will add the above explanation in the camera-ready version.
>
> Q: Mathematical definitions are not precise. To add to the writing issues, in many places the mathematical notation/assumptions are not properly defined. Some examples are given below:
>
> A: Thanks for the notification of the imprecise mathematical definitions. We will revise them in our camera-ready version.
>
> Q: No connection with personalization. The title of the paper states "Personalized One Shot Federated Learning". The authors also write about the benefits of personalization in paragraph 4 of the Introduction section. However, the rest of the paper just focuses on the server learning a single global model. I don't understand what is the connection with personalization here.
>
> A: Thanks for the comments on the aspect of the personalization. We understand that the personalized federated learning setting may contain the multiple aspects. In this paper, we mainly consider that each client has a different biased dataset, which is one of the personalized federated learning settings. We agree that our claim of the personalization is somehow weak. We will weaken the claim of the personalization in the camera-ready version.

---

> ### Author Response · Authors · 2023-11-20
> **Response to Reviewer dmu6 (2/5)**
>
> Q: The idea of formulating FL as a posterior inference problem is already well-known in [1] (Al Shedivat et al. 2020)
>
> A: To the best of our knowledge, we are the first to consider the posterior inference problem in one-shot scenario. Note that the approach [1] requires a lengthy burn-in period before conducting posterior inference, for instance, 400 rounds, and it updates global model parameters by modifying the covariance-aggregated local models. It means that the algorithm [1] necessarily requires multiple iterations and cannot be used in a one-shot scenario. In contrast, our method FedLPA only requires immediate variational inference after training the local model, ensuring higher flexibility and efficiency in one-shot scenario.
>
> Besides, in the algorithm [1], obtaining statistical information to compute local covariances is of low rank. In reality, it fails to acquire the posterior of the aggregated model and cannot perform variational inference on the aggregated model. However, our method yields full-rank covariances, and after employing an expectation approximation method for variational inference on the aggregated model, we can achieve a usable global posterior.
>
> Q:The idea of approximating the Hessian as the Fisher is also standard in Ritter et al. 2018.
>
> A: In both the domain of natural gradient optimization [1][2][3] and modeling output uncertainty in variational inference [4], using the Fisher approximation of the Hessian does not involve the issue of inverting covariance. However, in the context of federated learning, when performing variational inference on the aggregated model, the necessity of inverting covariance becomes unavoidable. To address this problem, we propose a novel algorithm that constructs a quadratic objective function. During aggregation, this algorithm directly trains the aggregated model using local covariances and expectations, thereby circumventing the need for inversion operations. Additionally, we provide a detailed convergence analysis of this method in Appendix F, labeled as "CONVERGENCE ANALYSIS OF EQ. 14".
>
> [1]M. Al-Shedivat, J. Gillenwater, E. Xing, and A. Rostamizadeh, “Federated learning via posterior averaging: A new perspective and practical algorithms.
>
> [2]James Martens and Roger Grosse. Optimizing neural networks with kronecker-factored approximate curvature. In International conference on machine learning.
>
> [3]Roger Grosse and James Martens. A kronecker-factored approximate fisher matrix for convolution layers. In International Conference on Machine Learning.
>
> [4]A. Botev, H. Ritter, and D. Barber, “Practical gauss-newton optimisation for deep learning,” in International Conference on Machine Learning.
>
> [5]H. Ritter, A. Botev, and D. Barber, “A scalable laplace approximation for neural networks,” in 6th International Conference on Learning Representations, ICLR 2018- Conference Track Proceedings.
>
> [6] Matena, Michael S., and Colin A. Raffel. "Merging models with fisher-weighted averaging." Advances in Neural Information Processing Systems 35 (2022): 17703-17716.
>
> [7]Liangxi Liu, Feng Zheng, Hong Chen, Guo-Jun Qi, Heng Huang, and Ling Shao. A bayesian federated learning framework with online laplace approximation. arXiv preprint arXiv:2102.01936, 2021.
>
> [8]James Martens and Roger Grosse. Optimizing neural networks with Kronecker-factored approximate curvature. In International conference on machine learning.
>
> [9]Roger Grosse and James Martens. A Kronecker-factored approximate fisher matrix for convolution layers. In International Conference on Machine Learning.
>
> [10]A. Botev, H. Ritter, and D. Barber, “Practical gauss-newton optimisation for deep learning,” in International Conference on Machine Learning.
>
> [11]Divyansh Jhunjhunwala, Shiqiang Wang, and Gauri Joshi. Towards a theoretical and practical understanding of one-shot federated learning with fisher information. In Federated Learning and Analytics in Practice: Algorithms, Systems, Applications, and Opportunities, 2023.

---

> ### Author Response · Authors · 2023-11-20
> **Response to Reviewer dmu6 (3/5)**
>
> Q: Moreover, the resulting concept of using the Fisher information to aggregate models has already been well-explored in previous work [6], which the authors have not cited. The only difference here is that FedLPA proposes to use the K-FAC while [6] uses the diagonal Fisher. However, I feel this is not significant enough novelty.
>
> A: Thank you for suggesting the reference material, which has helped enrich the related work section of our paper. We will incorporate this into our paper in the future and cite it accordingly. We would like to briefly mention that we have already referenced a similar method DiagonalFisher[7] that utilizes Online Laplace Approximation to obtain diagonal Fisher for model aggregation. Note that [6] and [7] adopts the same core approach that utilizes the diagonal Fisher to aggregate models, in which they conduct experiments on different dataset and published on different venue. Based on our understanding, we mainly analyse our approach with comparison of [7].
>
> In [7], Diagonal Fisher assumes independence among parameters, neglecting inter-parameter correlations, resulting in inaccurate posterior approximations. However, strong correlations exist among parameters within each layer, such as matching patterns in convolutional kernels within convolutional networks. This is a crucial factor that cannot be overlooked; otherwise, aggregation of the posterior would result in lower posterior regions, as compared to our method, as illustrated in Figure 6 of our paper. In complex environments, employing diagonal Fisher for aggregation would prove to be entirely ineffective, whereas our method effectively leverages inter-parameter correlations at each layer, rendering it more robust. To demonstrate, we present results comparing aggregation using diagonal Fisher and our method. We have added experiments using the settings our paper and an MLP model (784-256-64-10) on the FMNIST dataset with three random seeds for one-shot FL, the client number is 10 and the beta=0.01. The results are as follows:
>
>
>
> | **Initial** | **FedAvg**   | **FedProx**   |**SCAFFOLD**   |**DiagonalFisher**| **FedLPA**|
> |---------------|------------------------|------------------------|------------------------|------------------------|------------------------|
> | Same        |42.35±0.16|24.80±0.10|42.10±0.15|56.34±0.34|76.63±0.04|
> | Different    |10.00±0.00|24.12±0.02|10.16±0.70|10.51±0.11|73.73±0.07|
>
>
>
> In the table, "Initial" denotes whether the client models were initialized using the same parameter values or independently.
>
> When "Initial" is set to "Same", all client models are trained on their respective datasets using identical parameter values for initialization. Consequently, there exists strong correlation among the local models. Additionally, in this scenario, model aggregation is equivalent to aggregating updates of local models. Although DiagonalFisher performs reasonably well under this condition, our method demonstrates superior performance, exhibiting a 20.29% increase in global test accuracy.
>
> When "Initial" is set to "Different", the models on different clients start training with distinct parameter values. Due to the high heterogeneity of local datasets, there is minimal correlation among local models. In this extreme scenario, DiagonalFisher [7] completely fails, while our method maintains an accuracy of 73.73%, showcasing remarkable robustness.
>
> It is essential to consider the indispensability of parameter correlations, which is why we compute correlations among parameters within layers to ensure the robustness and accuracy of model aggregation.
>
> Now, we discuss some related works which directly utilize K-FAC to approximate fisher matrix  and make a comparison with our proposed approach FedLPA.  [8][9][10] have provided us with significant inspiration. However, methods like K-FAC do not require computing the inverse of covariance. Nevertheless, in the context of federated learning, the necessity of inverting covariance becomes unavoidable during variational inference on the aggregated model.
>
> Methods like K-FAC assume direct independence among data samples to utilize expectation approximation. They obtain the inverse of Fisher from individual samples and then directly compute the expectation, thereby avoiding inverse operations. However, the expectation approximation inevitably leads to biased results during model aggregation. Detailed analysis can be found in Appendix E.
>
> To address this issue, we propose a novel algorithm that constructs a quadratic objective function. During aggregation, this algorithm directly trains the aggregated model using local covariances and expectations, eliminating the need for inversion operations. This aims to minimize aggregation biases as much as possible, and we provide a comprehensive convergence analysis of this method in Appendix F.
>
> This approach signifies our novelty in addressing these challenges.

---

> ### Author Response · Authors · 2023-11-20
> **Response to Reviewer dmu6 (4/5)**
>
> Q：Moreover, implementing the K-FAC requires clients to send $A_k$, $B_k$, $M_k$ matrices which increases communication compared to just the diagonal Fisher.
>
> A：Although the number of uploaded bits increased per round, it resulted in a significant improvement in the final outcome. Additionally, the increase in transmitted bits enhanced the robustness of the aggregation method. Moreover, as indicated in Table 4 of the paper, we observe only a marginal increase in the amount of communication required.
>
> A fully-connected neural network model with architecture 784-254-64-10, has 784∗256+256+254∗64+64+64∗10+10 = 217930 floating point numbers, which is 6973760 bits or around 0.831 MB. For one communication from a client to the server, our approach needs to upload additional $A_k$ and $B_k$, which have 785∗785+256∗256+257∗257+64∗64+65∗65+10∗10 = 756231 floating point numbers. Note, $A_k$ and $B_k$ are symmetric matrices, so we only need to upload the upper triangular part of $A_k$ and $B_k$, which is around 756231/2 = 378115.5 floating point numbers and 1.442 MB. Therefore, our approach costs 2.272 MB for the one directed communication, which is only 1.367 times than DiagonalFisher while DiagonalFisher costs 0.831*2 = 1.662 MB. We show the following tables based on the previous experiment results .
>
>
> | **Method** | **Global Test Acc / MB**   |
> |---------------|------------------------|
> | DiagonalFisher  |56.34/(1.662*10) = 3.39|
> | FedLPA              |76.63/(2.272*10) = 3.37|
>
> When "Initial" is set to "Same", the efficiency of every bit is almost the same.
>
>
> | **Method** | **Global Test Acc / MB**   |
> |---------------|------------------------|
> | DiagonalFisher  |10.51/(1.662*10) = 0.63 |
> | FedLPA              |73.73/(2.272*10) = 3.25|
>
> When "Initial" is set to "Different", the efficiency of every bit for our method is much higher than the DiagonalFisher.
>
> We will add the above analysis in the camera-ready version.
>
>
> Q:The authors also claim without any justification that  $A_k$, $B_k$, $M_k$  preserve the data/label privacy. This claim has to be supported by empirical evidence/theory in order for it to be justified.
>
> A: Thanks for proposing the privacy concern. In the FedLPA, $A_k$ is computed via the activations while $B_k$ is computed via the linear pre-activations of the layer. We note that $A_k$, $B_k$ and $M_k$ do not carry any label information, thus the transmission of $A_k$, $B_k$ and $M_k$ will not leak any label privacy. Otherwise, we agree that  $A_k$, $B_k$ and $M_k$ are a function of data which may contain privacy-sentive information of the local training data. However, our proposed FedLPA has the same privacy-preserving level as the conventional federated learning algorithms (i.e, FedAvg, FedProx, FedNova and Dense), which are all vulnerable to some privacy attacks (e.g, membership inference [1] or reconstruction attacks [2]). Inspired by other reviewers comments, DP-based FedLPA is a promising future direction to enhance the privacy-preserving level of FedLPA. Due to the above analysis, we will revise the sentence into ‘which should preserve the label privacy for the clients.’, and we will add the privacy analysis into the draft in the camera-ready version.
>
>
> Q: Weak baselines and easy datasets. Among the baselines, I understand the comparison with FedAvg to show improvement over simple averaging. However SCAFFOLD, FedNova and FedProx are not good one-shot baselines. These methods are primarily designed for multi-round FL and therefore I am not surprised to see that their performance is similar to FedAvg in the experiments. DENSE is the only one-shot basline which can be considered competitive. The authors have cited other one-shot algorithms like FedOV and FedCAVE but have not compared them based on the argument that they entail sharing more client side information. However, I think it fair to compare these algorithms since FedLPA also requires clients to share and $A_k$ and $B_k$ matrices.
>
> A: We note that $A_k$, $B_k$ and $M_k$ will not carry the label information, thus, the transmission of $A_k$, $B_k$ and $M_k$ not leak any label privacy. And in our paper, we mention that “FedOV and FEDCAVE entail sharing more client-side label information or transmitting client label information to the server, which may jeopardize label privacy and are beyond the scope of this study”. As, FEDCVAE (Heinbaugh et al., 2022) needed all the client label distribution to be transmitted to the server side and FedOV (Diao et al. 2023) needed the clients to know the labels which were unknown [11]. In this case to compare them fairly, we do not choose the FedOV and FedCAVE as the baseline approach.
>
> Even so, in our appendix G.7 EXPERIMENTS WITH FEDOV, we show in some cases, our method FedLPA could outperform the state-of-the-art method FedOV. Note that FedOV consumes more computing resources, needs more label information, needs an extra distillation process and needs an extra outlier generation process.

---

> ### Author Response · Authors · 2023-11-20
> **Response to Reviewer dmu6 (5/5)**
>
> Q: FedLPA seems to outperform baselines significantly only for easier datasets such as FMNIST, MNIST and SVHN. For CIFAR-10 (which can be considered as the hardest dataset), the improvement over DENSE seems to reduce.
>
> A: In almost all the settings, our method can outperform the state-of-the-art baseline approach DENSE. In our extensive experiments with different local epoch numbers on all the four datasets, our method outperform DENSE in most cases.  We note that DENSE consumes more computing resources as we shown in Table 4 in the draft. Besides, it needs an extra data generation stage and an extra model distillation stage. Our method could get better results and consume less resources. What’s more, in Section 4.6, we also show that our method has the potential to extend to multiple-round settings, while it is hard to extend the DENSE into multi-round settings.
>
> Q: Therefore I was interested in seeing the performance of FedLPA for even harder datasets such as CIFAR-100 and Tiny-ImageNet as done in the DENSE paper. In addition, the paper would be significantly strengthened if the authors considered more realistic FL datasets such as EMNIST and Shakespeare.
>
> A: Thanks for the suggestion. We have added the experiment with our method on the same experiment setting in the paper with Resnet-18 with three random seeds with 10 clients. We conducted experiments on CIFAR-100 datasets with beta=0.1, 0.3 and 0.5. The results are shown as follows:
>
> | **Beta**     | **Accuracy**        |
> |---------------|------------------------|
> | **0.1**       | 15.11±0.38 |
> | **0.3**       | 18.82±0.71|
> | **0.5**       | 21.77±0.03 |
>
>
> We can see that even with the complicated dataset CIFAR-100, our method could also get satisfactory results in the federated one-shot setting.
>
> Besides, we also have added the experiments on EMNST using simple CNN with 10 clients and three random seeds. We do the experiments on EMNIST-mnist and EMNIST-letters.
>
> EMNIT-mnist (10 classes)
>
> | **Beta**     | **FedAvg**        | **FedLPA**|
> |---------------|------------------------|-------------|
> | **0.1**       | 57.63±2.30 | 74.23±3.10|
> | **0.3**       | 62.32±1.77| 86.55±0.24|
> | **0.5**       | 82.71±0.96 | 91.75±0.26|
>
>
> EMNIT-letters (37 classes)
>
> | **Beta**     | **FedAvg**        | **FedLPA**|
> |---------------|------------------------|-------------|
> | **0.1**       | 16.22±0.38 | 26.34±0.71|
> | **0.3**       | 25.51±0.44| 31.75±0.03|
> | **0.5**       | 26.34±0.07 | 33.78±0.14|
>
> We can see that our prosed approach FedLPA works well on the EMNIST dataset. Due to the rebuttal time limit, we have not added the experiments on Shakespeare dataset. However, we believe that our proposed FedLPA works well on the U-net, as the implementation of FedLPA is not limited to the model architecture.
>
> Q: In addition, I was curious to know why the computation cost of FedProx is higher than FedAvg and FedNova in Table 4.
>
> A: In our paper, we mainly adopt the most-cited non-iid FL benchmark (https://github.com/Xtra-Computing/NIID-Bench) to get the fair comparison of FedLPA and other baselines. The reason that why computation cost of FedProx is higher than FedAvg may be that the FedProx adds a L2 regular term to make local updates around the global mode, which adds more computing overhead. Besides, using the original codebase (https://github.com/litian96/FedProx) from FedPorx also consumes more time than FedAvg and FedNova, under the above non-iid FL benchmark. We will add the above analysis in the camera-ready version.

---

> ### Author Response · Authors · 2023-11-22
> **Looking forward to your feedbacks for our rebuttal**
>
> Dear Reviewer dmu6,
>
> We'd appreciate it if you could provide any feedback and add further comments based on our rebuttals . We are willing to provide further quick reponses. Thanks for your time and efforts!

---

### Official Review · Reviewer_qtZC · 2023-10-31

**Soundness:** 4 excellent
**Presentation:** 4 excellent
**Contribution:** 3 good
**Rating:** 5
**Confidence:** 3

**Summary:**

This paper aims to improve one-shot federated learning where the clients and the server are allowed to communicate only once. Instead of gradients, the authors propose FedLPA that computes the local posteriors in a layer-wise manner and aggregates them to form the global posterior, which is later used to update the global model. To this end, the authors approximate the posteriors with Laplacian approximation and empirical fisher matrices. An additional approximation with a multi-variate linear objective is proposed to estimate the global parameters and avoid the potential bias of Fisher matrices induced by the independence assumption that prior works have. The experimental results demonstrate superior performance over classical federated optimization and a one-shot FL baseline in various data skewness settings.

**Strengths:**

1. The paper is well-written and easy to follow. The problem targeted in this work is meaningful.

2. The idea is interesting and seems novel. Instead of noisy gradient accumulation, the proposed method considers a layer-wise aggregation that may reduce interference and provide a clearer supervision signal for the global model. The additional approximation is also interesting, though lacking a proper comparison and analysis.

3. The proposed method empirically performs well in many settings despite the limited amount of baselines and architectures.

**Weaknesses:**

1. My biggest concern is privacy risks. I doubt that communicating pre- and activations is more vulnerable to attacks, such as membership inference [1] or reconstruction attacks [2], as it contains more precise information in a layer-wise manner. For example, it is known that due to the sparsity of ReLU functions, gradient inversion attacks are less efficient in a large-batch case. However, the proposed method reveals both pre and after-activation values and may open up new attack possibilities. I am not saying the authors must solve the issue in this work, but I urge the authors to discuss the potential risks that the proposed method may introduce and solutions, e.g., differential privacy.

2. Despite the impressive numbers, the experiment settings seem limited. The author only considers a simple CNN network and a baseline tailored for one-shot federated learning.

[1] Luca Melis, Congzheng Song, Emiliano De Cristofaro, and Vitaly Shmatikov. Exploiting unintended feature leakage in collaborative learning. In 2019 IEEE symposium on security and privacy (SP), 2019.

[2] Jonas Geiping, Hartmut Bauermeister, Hannah Droge, and Michael Moeller. Inverting gradients: how easy is it to break privacy in federated learning? Advances in Neural Information Processing Systems (NeurIPS), 2020.

**Questions:**

1. The authors claim a personalized federated learning setting. How does the proposed method customize the model for different users?

2. Can the proposed method scale up to more complex networks, such as ResNet or U-net?

---

> ### Author Response · Authors · 2023-11-20
> **Response to Reviewer qtZC**
>
> We kindly thank the reviewer for their constructive and valuable feedback. We appreciate the time and effort you have dedicated to assessing our work and we are happy to address some of the concerns below. We sincerely hope you can read our responses and adjust the scores if appropriate.
>
> Q: My biggest concern is privacy risks. I doubt that communicating pre- and activations is more vulnerable to attacks, such as membership inference [1] or reconstruction attacks [2], as it contains more precise information in a layer-wise manner. For example, it is known that due to the sparsity of ReLU functions, gradient inversion attacks are less efficient in a large-batch case. However, the proposed method reveals both pre and after-activation values and may open up new attack possibilities. I am not saying the authors must solve the issue in this work, but I urge the authors to discuss the potential risks that the proposed method may introduce and solutions, e.g., differential privacy.
>
> A: Thanks for proposing the privacy concern. In the FedLPA, A is computed via the activations while B is computed via the linear pre-activations of the layer. We note that A, B and M do not carry any label information, thus the transmission of A, B and M will not leak any label privacy. Otherwise, we agree that A, B and M are a function of data which may contain privacy-sensitive information of the local training data. However, our proposed FedLPA has the same privacy-preserving level as the conventional federated learning algorithms (i.e, FedAvg, FedProx, FedNova and Dense), which are all vulnerable to some privacy attacks (e.g, membership inference [1] or reconstruction attacks [2]). We agree that combining differential privacy with FedLPA can enhance the privacy protection of the training data, which is the most promising future direction. However, in this paper, we mainly focus on the efficiency and accuracy of one-shot federated learning algorithms and propose FedLPA which outperforms the SOTA one-shot algorithms on the efficiency and accuracy.
>
> [1] Luca Melis, Congzheng Song, Emiliano De Cristofaro, and Vitaly Shmatikov. Exploiting unintended feature leakage in collaborative learning. In 2019 IEEE symposium on security and privacy (SP), 2019.
>
> [2] Jonas Geiping, Hartmut Bauermeister, Hannah Droge, and Michael Moeller. Inverting gradients: how easy is it to break privacy in federated learning? Advances in Neural Information Processing Systems (NeurIPS), 2020.
>
> Q: Despite the impressive numbers, the experiment settings seem limited. The author only considers a simple CNN network and a baseline tailored for one-shot federated learning. Can the proposed method scale up to more complex networks, such as ResNet or U-net?
>
> A: Thanks for the suggestion. We have added the experiments with FedLPA on the same experiments setting in the paper with resnet-18 with three random seeds on CIFAR-10 dataset. We set the parameters with beta=0.1, 0.3 and 0.5 with 10 clients. The results are shown as follows:
>
> | **Beta**     | **Resnet-18**        | **Simple-cnn** |
> |---------------|------------------------|-------------|
> | **0.1**       | 23.62±0.51 | 19.97±0.02|
> | **0.3**       | 27.43±0.04| 26.60±0.01|
> | **0.5**       | 31.70±0.14 | 24.20±0.02|
>
> From the above table, we can see that with the more complex NN structure, our proposed method FedLPA can get better accuracy when in the federated one-shot settings. We will add the above table and results in the Appendix in the camera-ready version. Due to the rebuttal time limit, we have not added the experiments on the U-net. However, we believe that our proposed FedLPA works well on the U-net, as the implementation of FedLPA is not limited to the model architecture.
>
> We also note that in our appendix G.7, we show in some cases, our proposed method FedLPA can outperform the state-of-the-art federated one-shot approach FedOV, while the FedOV requires more label information with consuming more computing resources.
>
> Q: The authors claim a personalized federated learning setting. How does the proposed method customize the model for different users?
>
> A: Thanks for the comments on the aspect of the personalization. We understand that the personalized federated learning setting may contain the multiple aspects. In this paper, we mainly consider that each client has a different biased dataset, which is one of the personalized federated learning settings. We agree that our claim of the personalization is somehow weak. We will weaken the claim of the personalization in the camera-ready version.

---

> > ### Comment · Reviewer_qtZC · 2023-11-21
> > **Response to the authors**
> >
> > I appreciate the authors' response and would like to comment on the following points.
> >
> > ----
> > > The transmission of A, B, and M will not leak any label privacy
> >
> > This might not be true. Similar to what the authors have shown in Figure 1, the activations/features tend to be clustered together, thus leaking additional label information and increasing the risk of reconstruction attacks. The proposed method may open up the possibility of different types of reconstruction attacks, such as class-wise, statistical matching, and layer-wise attacks.
> >
> > ----
> > >  However, our proposed FedLPA has the same privacy-preserving level as the conventional federated learning algorithms (i.e., FedAvg, FedProx, FedNova, and Dense)
> >
> > This might not be true, either. As pointed out by other reviewers (8qhC, yUMy) and my previous point, the proposed method communicates more information that could increase privacy leakage, while conventional methods communicate solely averaged class-agnostic gradients. Therefore, the privacy risks for these two kinds of methods are totally different.
> >
> > ----
> > > The additional experiments
> >
> > I appreciate the provided experiments, but the baselines are missing here.
> >
> > ----
> > > Claims about personalized federated learning
> >
> > As also mentioned by other reviewers (8Erm, dmu6), the proposed method has a weak link to personalized federated learning. I'd recommend that the authors remove the term and rephrase the paper accordingly.
> >
> > ----
> > Overall, the paper seems borderline to me. I still appreciate the novelty of the idea; however, after reading the response, my concern about the privacy risks still remains. This concern also appears in other reviews and might not be able to be solved in the current version. Therefore, I will lower my score for now but remain open to adjustment.

---

> ### Author Response · Authors · 2023-11-22
> **Response to Reviewer qtZC's futher comments-1**
>
> Thanks for your quick reply.
>
> >Q: This might not be true. Similar to what the authors have shown in Figure 1, the activations/features tend to be clustered together, thus leaking additional label information and increasing the risk of reconstruction attacks. The proposed method may open up the possibility of different types of reconstruction attacks, such as class-wise, statistical matching, and layer-wise attacks.
>
> A: We apologize that we made a writing error that on Page 5, B is not the value of the pre-action but rather the gradient of the pre-action, as shown in equations Eq.23 and Eq. 24.  We will revise this in our camera-ready version. This may lead you to a wrong understanding about the privacy concern.
>
> We emphasize that our proposed FedLPA is also a class-agnostic approach.
> Here, we would like to re-emphasize our privacy analysis. For the privacy of the training labels, e.g, label distribution on local node, our computations on A, B and M never utilize any information of the label distributions. As a comparison, FedCAVE, which transmits client label information to the server, require the training label distribution to do the distillation.
> Besides, our t-SNE illustration in Figure 1 shows the classification capability on global model, which can clearly separate the classes. However, our figures of the t-SNE illustrations on local models in Appendix G.2 shows that for the data belong to the same class, their t-SNE illustrations are erratically distributed on different local nodes. For instance, for node 2, its training data only has 3 classes while most of the training data locates in class 5. However, it is hard for the server to infer that label distribution, since the t-SNE illustration both on node 2 and other nodes also seems irregular.
>
> >Q: This might not be true, either. As pointed out by other reviewers (8qhC, yUMy) and my previous point, the proposed method communicates more information that could increase privacy leakage, while conventional methods communicate solely averaged class-agnostic gradients. Therefore, the privacy risks for these two kinds of methods are totally different.
>
> A: We admit that our approach FedLPA provides more information than FedAvg, However, the additional information we provide is the mean of each sample in each dimension, the mean of squares of each sample in each dimension, and the mean of square gradients. These solely marginally enriches the attack capability of several reconstruction attacks. We have added following analysis for the explanation on that. In this case, we aim to claim that our privacy-preserving level is similar to FedAvg.

---

> ### Author Response · Authors · 2023-11-22
> **Response to Reviewer qtZC's futher comments-2**
>
> > Further discussion of privacy concern Part 1
>
> For privacy attacks, let me start by assuming the simplest scenario where each client has only one sample, and the model comprises a single layer, such as an MLP.
>
> Let $y = W * x$, (x is n+1 dimensional, with the last dimension being a unit value 1)
>
> $g = Df(y)/Dx$ (where f is the loss function)
>
> In this case, $A = x* x^T$, $B = g*g^T$
>
> In this single-sample scenario, an attacker can directly obtain 'x' from the last column of A. With 'x' and 'W', the attacker can acquire the model's output. Furthermore, utilizing the Loss and 'g', it's possible to get the label information.
>
> FedAvg would also be vulnerable to a reconstruction attack in this scenario, allowing the attacker to obtain sample and label information.
>
> When each client has two samples ($x1, x2 \in Dataset$), then:
>
> $A = 1/2 * x_1 * x_1^T + 1/2 * x_2 * x_2^T$
>
> $B = 1/2 * g_1 * g_1^T + 1/2 * g_2 * g_2^T$
>
> The last column c of A equals $1/2 * x_1 + 1/2 * x_2$
>
> The diagonal elements d of A equal $1/2 * x_1^2 + 1/2 * x_2^2$
>
> In the case of these two samples, an attacker can utilize the information from A and B to get the two samples $x_1$ and $x_2$. Using the same methodology, they can also obtain $g_1$ and $g_2$. Consequently, the attacker can reverse-engineer the labels as well.
>
> FedAvg could also potentially succumb to a reconstruction attack in this scenario, providing the attacker with sample and label information, although the obtained information might be more ambiguous.
>
> When each client has three or more samples ($x \in Dataset $):
>
> $A = E_{x \in Dataset }(x*x^T)$
>
> $B = E_{x \in Dataset }(g*g^T)$
>
> In this situation, the last column c of A, $c = E_{x \in Dataset}(x)$ represents the average of the sample dataset, depicting the projection of the data distribution in the sample space on various coordinate axes. Furthermore, the diagonal elements of A ($E_{x \in Dataset}(x*x^T)$) offer the attacker statistical information about this local dataset.
>
> Generally, solely using the statistical information of these datasets cannot reconstruct the entire dataset. Similarly, it's not possible to obtain gradients for the output of each sample, thereby preventing the reconstruction of individual sample labels. The results obtained by using c and W to gather statistical label information are unreliable.
>
>
> Additionally, for structures such as CNNs and RNNs/LSTMs, the difficulty of attacks increases due to weight sharing.
>
> For CNNs, since convolutional kernels only accept local samples as input, information in A encompasses statistical information from all localities of the samples.
>
> For RNNs/LSTMs, information in A includes statistics of each word vector in a sentence. These network structures make it possible for attackers to fail even in single-sample scenarios.
>
> For multilayer MLPs, the information contained in the intermediate layer A is almost equivalent to the information encoded in the parameters of the BN (Batch Normalization) layer. The mean output of the BN layer is equivalent to the last column of A, whereas the variances differ between the BN layer and A's diagonal but both contain statistical information related to squared values.
>
> It's worth noting that the parameters acquired by the BN layer using the sliding-window average method are also frequently used during the computation of A and B, as mentioned in the paper[1].
>
> FedAvg provides model parameter values, the average of gradients, and BN layer parameters. Compared to FedAvg, the additional information we offer is actually limited to: the mean of each sample in each dimension, the mean of squares of each sample in each dimension, and the mean of square gradients. Utilizing this information, attacking becomes highly challenging when the number of samples exceeds three. Although we don't rule out the possibility of successful methods in practice due to the data's own correlations, the limitations are significant based on our analysis, and our security level is quite close to that of FedAvg.

---

> ### Author Response · Authors · 2023-11-22
> **Response to Reviewer qtZC's futher comments-3**
>
> > Further discussion of privacy concern Part 2
>
> We discuss two common attacks here:
>
> Inferring class representatives:
> i) Model inversion attacks [2] exploit the confidence information provided by machine learning applications or services. Our method does not provide confidence information, nor does it compute the information required for it. Therefore, our method's defense level against these attacks aligns with FedAvg's defense level.
>
> ii) Attacks using GANs to construct class representatives [3] utilize the client-uploaded model as a discriminator and its output as labels to train a generator to generate similar data. The additional statistical information we provide might be used to constrain the distribution of inputs for GANs, specifically their mean values. Since the statistical information of the dataset may contain some common features among samples, it might potentially aid in speeding up the convergence of training GANs but may not significantly enhance the accuracy of generated data after GAN optimization. It's worth noting that if the BN layer parameters uploaded by FedAvg could be used to constrain the statistical information of GANs' inputs, they would be equivalent to the information provided by our method.
>
> Additionally, these attack methods against FedAvg only yield favorable results when class members are similar, meaning the dataset has clear common features that allow the constructed representatives to resemble the training data. When class members are dissimilar, these shared features tend to be confounded, rendering the constraints imposed by the sample mean ineffective, hence not enhancing the effectiveness of GANs attacks.
>
> In summary, our method exhibits a security level consistent with FedAvg against these types of attacks. Even in cases where the BN layer is not required, our method's security is similar to that of FedAvg.
>
> Membership inference attacks against aggregate statistics [2][3] and Membership inference attacks against ML models[4][5][6][7] aim to infer whether a sample belongs to the training dataset using appropriate prior distributions and statistical data. These attack methods impose specific requirements on the dataset. In such attack scenarios, whether the sample mean information our method can provide is exploitable by the attacker depends on whether this information can reveal the inherent distribution correlations within the dataset. However, for high-dimensional complex data, sample mean information often falls short in achieving this.
>
> Therefore, in the case of these attacks, our method exhibits the same level of security as FedAvg (since FedAvg requires uploading statistically equivalent information within the BN layer). For scenarios without a BN layer, whether our method reduces security depends on the characteristics of the dataset itself. Real-world data is often high-dimensional and complex, making successful attacks challenging.
>
> [1]Martens, James. Second-order optimization for neural networks. University of Toronto (Canada), 2016.
>
> [2]Fredrikson, Matt, Somesh Jha, and Thomas Ristenpart. "Model inversion attacks that exploit confidence information and basic countermeasures." Proceedings of the 22nd ACM SIGSAC conference on computer and communications security. 2015.
>
> [3]Hitaj, Briland, Giuseppe Ateniese, and Fernando Perez-Cruz. "Deep models under the GAN: information leakage from collaborative deep learning." Proceedings of the 2017 ACM SIGSAC conference on computer and communications security. 2017.
>
> [4] C. Dwork, A. Smith, T. Steinke, J. Ullman, and S. Vadhan. Robust traceability from trace amounts. In FOCS, 2015.
>
> [5] A. Pyrgelis, C. Troncoso, and E. De Cristofaro. Knock knock, who’s there? membership inference on aggregate location data. In NDSS, 2018.
>
> [6] J. Hayes, L. Melis, G. Danezis, and E. De Cristofaro. LOGAN: Membership inference attacks against generative models. In PETS, 2019.
>
> [7] Y. Long, V. Bindschaedler, L. Wang, D. Bu, X. Wang, H. Tang, C. A. Gunter, and K. Chen. Understanding membership inferences on well-generalized learning models. arXiv:1802.04889, 2018.
>
> [8] R. Shokri, M. Stronati, C. Song, and V. Shmatikov. Membership inference attacks against machine learning models. In S&P, 2017.
>
> [9] S. Truex, L. Liu, M. E. Gursoy, L. Yu, and W. Wei. Towards demystifying membership inference attacks. arXiv:1807.09173, 2018.

---

> ### Author Response · Authors · 2023-11-22
> **Response to Reviewer qtZC's futher comments-4**
>
> >Q: I appreciate the provided experiments, but the baselines are missing here.
>
> A: Thanks for the comments. Due to rebuttal time limit, we have not finished the baseline approaches on the ResNet-18. We will comment this back once completed.
>
> >Q: As also mentioned by other reviewers (8Erm, dmu6), the proposed method has a weak link to personalized federated learning. I'd recommend that the authors remove the term and rephrase the paper accordingly.
>
> A: Thanks for the comments. We will remove the claim of the personalization in the next version, since it is a weak setting in our scheme.

---

> ### Author Response · Authors · 2023-11-22
> **Looking forward to your feedbacks for our rebuttal**
>
> Dear Reviewer qtZC,
>
> We'd appreciate it if you could provide any feedback and add further comments based on our rebuttals . We are willing to provide further quick reponses. Thanks for your time and efforts!

---

### Official Review · Reviewer_8qhC · 2023-11-01

**Soundness:** 2 fair
**Presentation:** 3 good
**Contribution:** 2 fair
**Rating:** 5
**Confidence:** 3

**Summary:**

The paper studies personalized one-shot federated learning (FL) where final global model is constructed through a one-shot update from the clients' local models. Since heterogeneous data makes one-shot FL challenging, the authors propose a Layer-wise Posterior Aggregation strategy called FedLPA. One of the main motivations behind this strategy is to improve the performance under non-iid data distribution and avoid leaking clients' sensitive data such as the label distribution.

**Strengths:**

The proposed approach, FedLPA, based on layer-wise posteriors does not increase the computational cost due to the block-diagonality assumption in the empirical Fisher information matrix. Empirically, FedLPA seems to outperform the baselines consistently and significantly.

**Weaknesses:**

The paper claims several times that FedLPA prevents additional leakage of user data at the one-shot step. Specifically, the claim is that the server only receives A, B, and M without any auxiliary information -- which should preserve the data/label privacy for the clients. However, A, B, and M do carry information about the client data since A and B are found from the empirical Fisher information matrix which is a function of the dataset and the locally trained model. So, if I am not missing something, A, B, and M are already a function of both the data and the labels. Then, how is this claim about privacy valid?

**Questions:**

Can you please clarify what the authors mean by this sentence (and similar ones in the paper) "The transmitted data between the clients and the server is solely $A_k$, $B_k$, $M_k$ without any extra auxiliary information, which preserves the data/label
privacy for the local clients."? Given that  $A_k$, $B_k$, $M_k$ are found via data-dependent empirical Fisher information matrix, how does FedLPA preserve data/label privacy?

---

> ### Author Response · Authors · 2023-11-20
> **Response to Reviewer 8qhC**
>
> We kindly thank the reviewer for their constructive and valuable feedback. We appreciate the time and effort you have dedicated to assessing our work and we are happy to address the concerns below. We sincerely hope you can read our responses and adjust the scores if appropriate.
>
> Q: The paper claims several times that FedLPA prevents additional leakage of user data at the one-shot step. Specifically, the claim is that the server only receives $A_k$, $B_k$ and $M_k$ without any auxiliary information -- which should preserve the data/label privacy for the clients. However, $A_k$, $B_k$ and $M_k$ do carry information about the client data since A and B are found from the empirical Fisher information matrix which is a function of the dataset and the locally trained model. So, if I am not missing something, $A_k$, $B_k$ and $M_k$ are already a function of both the data and the labels. Then, how is this claim about privacy valid?
>
>
> A: Thanks for proposing the privacy concern. In the FedLPA, $A_k$ is computed via the activations while $B_k$ is computed via the linear pre-activations of the layer. We note that $A_k$, $B_k$ and $M_k$ do not carry any label information, thus the transmission of $A_k$, $B_k$ and $M_k$ will not leak any label privacy. Otherwise, we agree that $A_k$, $B_k$ and $M_k$  are a function of data which may contain privacy-sensitive information of the local training data. However, our proposed FedLPA has the same privacy-preserving level as the conventional federated learning algorithms (i.e, FedAvg, FedProx, FedNova and Dense), which are all vulnerable to some privacy attacks (e.g, membership inference [1] or reconstruction attacks [2]). Inspired by other reviewers’ comments, DP-based FedLPA is a promising future direction to enhance the privacy-preserving level of FedLPA. Due to the above analysis, we will revise the sentence into “which should preserve the label privacy for the clients”, and we will add the privacy analysis into the draft in the camera-ready version.
>
> [1] Luca Melis, Congzheng Song, Emiliano De Cristofaro, and Vitaly Shmatikov. Exploiting unintended feature leakage in collaborative learning. In 2019 IEEE symposium on security and privacy (SP), 2019.
>
> [2] Jonas Geiping, Hartmut Bauermeister, Hannah Droge, and Michael Moeller. Inverting gradients: how easy is it to break privacy in federated learning? Advances in Neural Information Processing Systems (NeurIPS), 2020.

---

> > ### Comment · Reviewer_8qhC · 2023-11-22
> > **response to the rebuttal**
> >
> > I thank the authors for the explanation regarding the privacy claims. However, I am not convinced that the proposed method provides any privacy gains. The release of additional matrices increases the privacy leakage from the training data (inputs) even if it doesn't contain any information from the labels (outputs). I am also not sure if "label privacy for the clients" is a meaningful notion of privacy as both the training data (inputs) and the labels are user data.

---

> ### Author Response · Authors · 2023-11-22
> **Response to Reviewer qtZC's futher comments**
>
> >Q: I thank the authors for the explanation regarding the privacy claims. However, I am not convinced that the proposed method provides any privacy gains. The release of additional matrices increases the privacy leakage from the training data (inputs) even if it doesn't contain any information from the labels (outputs). I am also not sure if "label privacy for the clients" is a meaningful notion of privacy as both the training data (inputs) and the labels are user data.
>
> A: Thanks for your quick reply. We have added a [common response](https://openreview.net/forum?id=WroPkTLiAJ&noteId=rpQph8X8cd) about the training data privacy protection level in our scheme. In summary, our approach FedLPA protects the label privacy (e.g., training label distribution) and has a similar privacy protection level on training data privacy, compared to FedAvg.
>
> For the privacy of the training labels, e.g, label distribution on local node, our computations on A, B and M never utilize any information of the label distributions. As a comparison, FedCAVE, whic transmits client label information to the server, require the training label distribution to do the distillation.
>
> Several papers [1][2] have notified that the label privacy, e.g., the concern of label distribution leakage and raw label leakage, is sensitive in federated learning. We believe that it has also been a concern in fed one-shot scenario.
>
> For example, in Appendix G.2, we show that node 2 solely has 3 classes, in which most of training data locates in class 5. This label distribution information is sensitive, as the node do not want other node or the server to know it.
>
> [1] Sun, Jiankai, et al. "Label leakage and protection from forward embedding in vertical federated learning." arXiv preprint arXiv:2203.01451 (2022).
>
> [2] Wainakh, Aidmar, et al. "User-level label leakage from gradients in federated learning." arXiv preprint arXiv:2105.09369 (2021).

---

> ### Author Response · Authors · 2023-11-22
> **Looking forward to your feedbacks for our rebuttal**
>
> Dear Reviewer 8qhC,
>
> We'd appreciate it if you could provide any feedback and add further comments based on our rebuttals . We are willing to provide further quick reponses. Thanks for your time and efforts!

---

### Author Response · Authors · 2023-11-22
**General Responses to Privacy Concerns**

Thanks for the reviewers’ quick replies. Here, we want to add a revised general response to all reviewers about the privacy concern, especially on the training data and label privacy.  In summary, our approach FedLPA protects the label privacy (e.g., training label distribution) and has a similar privacy protection level on training data privacy, compared to FedAvg.
For the privacy of the training labels, e.g, label distribution on local node, our computations on A, B and M never utilize any information of the label distributions. As a comparison, FedCAVE, whic transmits client label information to the server, require the training label distribution to do the distillation.
For the training data privacy, we have added some analysis. Please refer to [Response to Reviewer qtZC's futher comments-2](https://openreview.net/forum?id=WroPkTLiAJ&noteId=MY3WC3vJxm) and to [Response to Reviewer qtZC's futher comments-3]( https://openreview.net/forum?id=WroPkTLiAJ&noteId=Bd35ZmKcLk).

---

### Meta-Review · Area_Chair_ghde · 2023-12-14

**Metareview:**

This paper introduces a one-shot federated learning approach under a Bayesian framework. The goal is to find a sufficiently concise approximation of the individual posterior distributions on the model params to enable efficient communication (without much loss in utility). Using a sequence of known approximation methods applied layer-wise the proposed scheme significantly reduce the communication cost  The authors compare the technique with baselines show that the proposed FedLPA outperformed those methods in a one-shot Federated Learning with a synthetic partition into small number of clients. The approach seems to hold promise but the reviewers pointed out a number of issues that jointly suggest that the work is not quite ready for publication. Most notably, the paper does not have any concrete FL scenario to evaluate all the claims and design choices against. Without some additional specifics it's impossible to evaluate whether the additional loss of privacy is reasonable, whether the synthetic data partition is meaningful and whether one global model (vs a collection of personalized ones) is a good metric. This seems to result from the authors targeting abstract generality but also makes it hard to evaluate the practical significance and relevance of the multitute of design and evaluation choices.

**Justification For Why Not Higher Score:**

as explained

**Justification For Why Not Lower Score:**

n/a

---

### Decision · Program_Chairs · 2024-01-16

Reject